# ATTENTION-AWARE POST-TRAINING QUANTIZATION WITHOUT BACKPROPAGATION

## ABSTRACT

Quantization offers a promising solution for deploying large-scale language models (LLMs) on resource-constrained devices. However, early quantization methods, developed for smaller networks like ResNet, rely on gradient-based optimization, which becomes impractical for hyper-scale LLMs with billions of parameters. While recently proposed backpropagation-free post-training quantization (PTQ) methods alleviate this issue, their performance is limited by a lack of inter-layer dependency consideration. In this paper, we introduce a novel PTQ algorithm that incorporates inter-layer dependencies without relying on backpropagation. The key innovation is the development of attention-aware Hessian matrices that capture inter-layer interactions within the attention module. Extensive experiments demonstrate that our approach significantly outperforms conventional PTQ methods, particularly at low bit-widths.

## 1 INTRODUCTION

The explosive growth in complexity (parameters) of large-scale language models (LLMs) based on Transformers (Touvron et al., 2023; Zhang et al., 2022) has resulted in a proportional increase in computational costs, which has prompted an urgent need for efficient model processing and compression strategies. Quantization has emerged as a pivotal solution in this context, and it serves as an essential step in deploying AI models on resource-constrained devices that primarily support fixed-point arithmetic. By reducing precision, the memory bandwidth requirements can be alleviated, and the significant parallelism of quantized models can be SIMDified using highly efficient vector processing units, such as neural processing units (NPUs).

Two main categories of quantization approaches have been proposed to preserve the performance of original full-precision models: quantization-aware training (QAT) and post-training quantization (PTQ). Although QAT can potentially outperform PTQ, its practicality diminishes considerably when handling hyper-scale LLMs featuring billions of parameters. Consequently, recent quantization efforts have been directed toward PTQ.

Although classic PTQ methods have successfully quantized small-scale models (Nagel et al., 2020; Li et al., 2021), they rely on time-consuming gradient-based optimization, so their efficacy decreases when the complexity of LLMs increases. Accordingly, backpropagation-free PTQ methods have been developed for LLMs (Frantar et al., 2023; Xiao et al., 2023; Jeon et al., 2023b); however, their performance is somewhat limited owing to the lack of consideration of inter-layer dependencies. Recent studies have attempted to consider inter-layer dependencies (Shao et al., 2023; Ma et al., 2024), but they still rely on time-consuming gradient-based optimizations.

In this paper, we propose a novel quantization algorithm that considers inter-layer dependencies without relying on backpropagation. Our primary contributions can be summarized as follows:

- We propose a novel PTQ algorithm called BOA[1]. To avoid time-consuming gradient-based optimization, we adopt the Hessian-based strategy introduced by (Frantar & Alistarh, 2022). The primary contribution is to approximate the Hessian more accurately by exploiting the attention reconstruction error, not the layer-wise reconstruction error, to capture inter-layer dependencies within the attention module (**Section 3.2**).

---

[1]BOA: Backpropagation-free optimization for Attention-aware PTQ

- While the proposed Hessian facilitates the consideration of inter-layer dependencies, it requires a large amount of memory and high computational cost. Therefore, we incorporate several techniques to mitigate the computational overhead, including Hessian relaxation, efficient computation of inverse Hessians, and head-wise simultaneous quantization (**Section 3.3**).

- We evaluate BOA via extensive experiments on publicly available LLMs. Our results demonstrate that BOA outperforms conventional LLM PTQ methods by a significant margin, particularly for low-bit precision (*e.g.*, INT2) (**Section 4**).

## 2 RELATED WORKS

When calibration data are available, PTQ primarily aims to minimize the increase in task loss incurred by quantization. Consider a neural network parameterized by weights $\mathbf{W}$. Provided that the network is trained to convergence, the problem of quantizing $\mathbf{W}$ to minimize task loss degradation can be formulated as (LeCun et al., 1989)

$$\min_{\Delta \mathbf{w}} \ \Delta \mathbf{w}^T \cdot \mathbf{H}^{(\mathbf{w})} \cdot \Delta \mathbf{w}, \tag{1}$$

where $\mathbf{H}^{(\mathbf{w})}$ is the Hessian related to the flattened weight $\mathbf{w}$ and $\Delta \mathbf{w}$ is a weight perturbation caused by the quantization. Owing to the infeasibility in computing and storing the exact Hessian $\mathbf{H}^{(\mathbf{w})}$, many studies have assumed independence between layers, which relaxes (1) into the following layer-wise reconstruction problem (Nagel et al., 2020):

$$\min_{\Delta \mathbf{W}^{(\ell)}} \ \left\| \Delta \mathbf{W}^{(\ell)} \mathbf{X}^{(\ell-1)} \right\|_F^2, \tag{2}$$

where $\mathbf{X}^{(\ell-1)}$ is the input to the $\ell$-th layer parameterized by $\mathbf{W}^{(\ell)}$.

To solve (2), early efforts aimed to optimize the weight-rounding mechanism (Nagel et al., 2020; Hubara et al., 2021; Li et al., 2021; Jeon et al., 2022; 2023a). Instead of allocating the nearest quantization bin, these studies attempted to assign quantized values that minimize the reconstruction error. In (Nagel et al., 2020), a backpropagation-based optimization method, called AdaRound, has been proposed. This algorithm has been extended to BRECQ where the block-wise reconstruction error has been used instead of the layer-wise reconstruction error to consider the inter-layer dependencies within a certain network block (*e.g.*, Transformer block), which leads to an enhanced low-bit quantization performance (Li et al., 2021). Although AdaRound and BRECQ have successfully quantized small-sized models such as ResNet (He et al., 2016), they are heavily dependent on time-consuming gradient-based optimizations. This renders their application to LLMs with billions of parameters challenging. Consequently, recent efforts have shifted towards the development of cost-effective quantization methods for LLMs.

These efforts can be classified into two orthogonal categories: 1) Hessian-based methods that optimize a weight-rounding mechanism without relying on backpropagation (*e.g.*, GPTQ (Frantar et al., 2023)) and 2) equivalent transformation (ET)-based methods that transform a model to be robust to quantization, thereby enhancing the performance of the naive rounding-to-nearest quantization (*e.g.*, SmoothQuant (Xiao et al., 2023), AWQ (Lin et al., 2024), Z-FOLD (Jeon et al., 2023b), Outlier Suppression+ (OS+) (Wei et al., 2023), OmniQuant (Shao et al., 2023), AffineQuant (Ma et al., 2024)). Among them, GPTQ has emerged as one of the most efficient quantization methods; GPTQ is capable of quantizing models with over 30 billion parameters in just a few GPU hours, with negligible performance degradation at INT4 precision. Moreover, GPTQ can be integrated with ET-based methods to enhance their performance (Lin et al., 2024; Jeon et al., 2023b).[2] Similar to GPTQ, we optimize a weight-rounding mechanism based on Hessian without relying on backpropagation. The primary difference is that we pursue the preservation of the attention output after the quantization while GPTQ aims to preserve each layer output and thus cannot consider inter-layer dependencies within the attention module.

Other algorithms exploiting different quantization strategies have also been proposed. For example, SpQR (Dettmers et al., 2023), SqueezeLLM (Kim et al., 2023), and OAC (Edalati et al., 2024) tried

---

[2] By transforming models with ET-based methods and then optimizing a weight-rounding mechanism with GPTQ (instead of applying the rounding-to-nearest method), the quantization performance can be boosted.

to preserve quantization-sensitive weights by assigning a large bit-width or retaining them in full-precision. QuIP (Chee et al., 2023) introduced the idea of incoherent processing to suppress outliers within weights. When compared to the standard uniform quantization, these algorithms require additional processing and memory costs in the real inference stage and need some special hardware and dedicated kernels without which accelerating the inference process may not be easy.[3] Furthermore, unlike server-grade GPUs (*e.g.* NVIDIA A100), on-device NPUs (*e.g.* Qualcomm Hexagon) lack support for the mixed precision format and such additional processing, and customizing kernels for desired functionalities is very challenging on on-device NPUs. Thus, we exclude these algorithms in our comparison and focus on the more universally supported uniform quantization format.

## 3 METHOD

### 3.1 OVERVIEW OF PROPOSED BOA

Similar to GPTQ, the proposed BOA algorithm quantizes weights by repeating the quantization and weight-update steps; once BOA quantizes one weight, it updates the remaining (not-yet-quantized) weights by exploiting the Hessian-based weight-update formula introduced by GPTQ, compensating for the task loss degradation caused by the quantization. When the $q$-th weight $w_q$ is quantized, the weight-update $\boldsymbol{\delta w}$ is mathematically expressed as

$$\boldsymbol{\delta w} = -\frac{w_q - \mathcal{Q}(w_q)}{[\mathbf{U}]_{q,q}}[\mathbf{U}]_{q,:} \text{ where } \mathbf{U} = \text{Chol}(\mathbf{H}^{-1})^T. \tag{3}$$

Here, $\mathbf{H}$ is the Hessian, $\text{Chol}(\cdot)$ denotes a Cholesky decomposition (*i.e.*, $\mathbf{U}$ is an upper triangular matrix satisfying $\mathbf{H}^{-1} = \mathbf{U}^T\mathbf{U}$), and $\mathcal{Q}$ is a uniform quantization function defined as

$$\mathcal{Q}(x) = s\left(\text{clamp}\left(\left\lfloor\frac{x}{s}\right\rceil + z, 0, 2^n - 1\right) - z\right),$$

where $s, z, n$ are the scale, zero-point, and bit-width, respectively, and $\lfloor\cdot\rceil$ represents the round-off.

The key difference over GPTQ lies in the approximation of the Hessian $\mathbf{H}$. In GPTQ, the layer-wise independence has been assumed to approximate $\mathbf{H}$, which yields the following Hessian equation[4]:

$$\mathbf{H}^{(\boldsymbol{w}^{(\ell)})} \approx 2\mathbf{X}^{(\ell-1)}\mathbf{X}^{(\ell-1)^T} \otimes \mathbf{I}, \tag{4}$$

where $\mathbf{H}^{(\boldsymbol{w}^{(\ell)})}$ is the Hessian for the $\ell$-th layer, $\otimes$ denotes the Kronecker product operation and $\mathbf{I}$ is the identity matrix. As the approximated Hessian $\mathbf{H}^{(\boldsymbol{w}^{(\ell)})}$ relies solely on the input, GPTQ is unable to account for the influence of other layers when compensating for quantization error (see (3)). In other words, GPTQ neglects inter-layer dependencies within the attention module, a crucial aspect of Transformers (Vaswani et al., 2017), which results in somewhat constrained performance for low precision (e.g., INT2) (Jeon et al., 2023b). To overcome this, we develop Hessians that incorporate inter-layer dependencies within the attention module and then use them instead of the conventional Hessian in (4).

### 3.2 PROPOSED ATTENTION-AWARE HESSIAN

To consider the inter-layer dependencies within the attention module, we exploit the attention reconstruction error rather than the layer-wise reconstruction error when approximating the Hessian.

For an input sequence $\mathbf{X} \in \mathbb{R}^{d \times L}$, the output of the multi-head attention (MHA) is expressed as

$$\text{MHA}(\mathbf{X}) = \sum_{h=1}^{H} \mathbf{W}_{\text{out},h}(\mathbf{A}_h\mathbf{V}_h)^T, \quad \mathbf{A}_h = \sigma\left(\frac{\mathbf{Q}_h\mathbf{K}_h^T}{\sqrt{d_h}}\right), \tag{5}$$

---

[3]In QuIP, the weight $\mathbf{W}$ is multiplied by random orthogonal matrices $\mathbf{U}$ and $\mathbf{V}$ (*i.e.*, $\mathbf{W} \leftarrow \mathbf{U}\mathbf{W}\mathbf{V}^T$). While this incoherent processing can suppress outliers within weights, additional processing is needed to recover quantized weights (*i.e.*, $\widehat{\mathbf{W}} \leftarrow \mathbf{U}^T\widehat{\mathbf{W}}\mathbf{V}$; see Algorithm 2 in (Chee et al., 2023)). Such processing must be done during the inference, incurring additional inference time and memory costs for storing $\mathbf{U}$ and $\mathbf{V}$.

[4]For any $\mathbf{M}_1$ and $\mathbf{M}_2$, the second-order derivative of $\|\mathbf{M}_1\Delta\mathbf{W}\mathbf{M}_2\|_F^2$ with respect to $\Delta\mathbf{w}$ is $2\mathbf{M}_2\mathbf{M}_2^T \otimes \mathbf{M}_1^T\mathbf{M}_1$ (see Appendix A for the proof).

---

**Algorithm 1** BOA

---

**Input**: weights $\mathbf{W} \in \mathbb{R}^{d_{\text{row}} \times d_{\text{col}}}$ and inputs $\mathbf{X}$ of the Transformer layer

1: **def** BOA($\mathbf{W}, \mathbf{X}$)
2:     Initialize quantized output: $\mathbf{Q} \leftarrow \mathbf{0}_{H \times d_{\text{row}}/H \times d_{\text{col}}}$
3:     Initialize (row-wise) quantization errors: $\mathbf{E} \leftarrow \mathbf{0}_{H \times d_{\text{col}}}$
4:     Compute attention-aware Hessians: $\mathbf{H}_h = \mathbf{H}_{\text{col},h} \otimes \mathbf{H}_{\text{row},h}$       ▷ See Table 1
5:     Set step size (scale) $\mathbf{S}$: $\min_{\mathbf{S}} \text{tr}\left(\Delta\mathbf{W}\mathbf{H}_{\text{col},h}\Delta\mathbf{W}^T\right)$
6:     Compute inverse Hessians $\mathbf{H}_{\text{col},h}^{-1}$ and $\mathbf{H}_{\text{row},h}^{-1}$
7:     Compute $\mathbf{U}_{\text{col},h} = \text{Chol}(\mathbf{H}_{\text{col},h}^{-1})^T$ and $\mathbf{U}_{\text{row},h} = \text{Chol}(\mathbf{H}_{\text{row},h}^{-1})^T$
8:     **for** $j = 0, \ldots, d_{\text{row}}/H - 1$ **do**
9:         Construct $\mathbf{W}^{(j)} \in \mathbb{R}^{H \times d_{\text{col}}}$ by stacking the $j$-th rows $[\mathbf{W}_h]_{j,:}$
10:         Quantize $\mathbf{W}^{(j)}$: $(\mathbf{Q}_{:,j,:}, \mathbf{E}) \leftarrow \text{GPTQ}(\mathbf{W}^{(j)}, \mathbf{U}_{\text{col},h}, \mathbf{S})$     ▷ See Appendix D
11:         Update remaining rows: $[\mathbf{W}_h]_{j:,:} \leftarrow [\mathbf{W}_h]_{j:,:} - \frac{[\mathbf{U}_{\text{row},h}^T]_{j:,j} \cdot \mathbf{E}_{h,:} \cdot \mathbf{U}_{\text{col},h}}{[\mathbf{U}_{\text{row},h}]_{j,j}}$
12:     **end for**

**Output**: quantized weights $\mathbf{Q}$

---

where $\sigma$ is the row-wise softmax function, $H$ is the number of attention heads, $d_h$ is the embedding dimension of the $h$-th attention head, $[\mathbf{Q}_h | \mathbf{K}_h | \mathbf{V}_h] = \mathbf{X}^T[\mathbf{W}_{Q,h}^T | \mathbf{W}_{K,h}^T | \mathbf{W}_{V,h}^T]$, $\mathbf{W}_{\{Q,K,V\},h} \in \mathbb{R}^{d_h \times d}$, and $\mathbf{W}_{\text{out},h} \in \mathbb{R}^{d \times d_h}$.

**Hessian for $\mathbf{W}_{Q,h}$**  When $\mathbf{W}_{Q,h}$ is quantized, $\mathbf{W}_{\text{out},h}$ and $\mathbf{V}_h$ in (5) remain unchanged, but the attention weight $\mathbf{A}_h$ changes. Using the first-order Taylor polynomial, the perturbation in $\mathbf{A}_h$ can be approximated as

$$\Delta\mathbf{A}_h = \sigma\left(\frac{(\mathbf{Q}_h + \Delta\mathbf{Q}_h)\mathbf{K}_h^T}{\sqrt{d_h}}\right) - \sigma\left(\frac{\mathbf{Q}_h\mathbf{K}_h^T}{\sqrt{d_h}}\right) \approx \frac{\Delta\mathbf{Q}_h\mathbf{K}_h^T}{\sqrt{d_h}}\mathbf{J}_\sigma^T = \frac{\mathbf{X}^T\Delta\mathbf{W}_{Q,h}^T\mathbf{K}_h^T\mathbf{J}_\sigma^T}{\sqrt{d_h}}, \quad (6)$$

where $\mathbf{J}_\sigma$ is the Jacobian matrix of the softmax function $\sigma$. Thus, the attention reconstruction error is expressed as

$$\|\Delta\text{MHA}(\mathbf{X})\|_F^2 = \|\mathbf{W}_{\text{out},h}(\Delta\mathbf{A}_h\mathbf{V}_h)^T\|_F^2 = \left\|\frac{\mathbf{W}_{\text{out},h}\mathbf{V}_h^T\mathbf{J}_\sigma\mathbf{K}_h}{\sqrt{d_h}}\Delta\mathbf{W}_{Q,h}\mathbf{X}\right\|_F^2. \quad (7)$$

Combining this with Footnote 4 yields the following Hessian for $\mathbf{W}_{Q,h}$:

$$\mathbf{H}^{(\mathbf{w}_{Q,h})} = 2\mathbf{X}\mathbf{X}^T \otimes \frac{\mathbf{K}_h^T\mathbf{J}_\sigma^T\mathbf{V}_h\mathbf{W}_{\text{out},h}^T\mathbf{W}_{\text{out},h}\mathbf{V}_h^T\mathbf{J}_\sigma\mathbf{K}_h}{d_h}. \quad (8)$$

**Hessian for $\mathbf{W}_{K,h}$**  When $\mathbf{W}_{K,h}$ is quantized, the attention weight $\mathbf{A}_h$ changes as in the quantization of $\mathbf{W}_{Q,h}$. By following the steps for (6), $\mathbf{A}_h$ can be approximated as

$$\Delta\mathbf{A}_h \approx \frac{\mathbf{Q}_h\Delta\mathbf{K}_h^T}{\sqrt{d_h}}\mathbf{J}_\sigma^T = \frac{\mathbf{Q}_h\Delta\mathbf{W}_{K,h}\mathbf{X}\mathbf{J}_\sigma^T}{\sqrt{d_h}}, \quad (9)$$

and thus the attention reconstruction error can be expressed as

$$\|\Delta\text{MHA}(\mathbf{X})\|_F^2 = \left\|\Delta\mathbf{A}_h\mathbf{V}_h\mathbf{W}_{\text{out},h}^T\right\|_F^2 = \left\|\frac{\mathbf{Q}_h}{\sqrt{d_h}}\Delta\mathbf{W}_{K,h}\mathbf{X}\mathbf{J}_\sigma^T\mathbf{V}_h\mathbf{W}_{\text{out},h}^T\right\|_F^2. \quad (10)$$

Thus, we obtain the following Hessian for $\mathbf{W}_{K,h}$:

$$\mathbf{H}^{(\mathbf{w}_{K,h})} = 2\mathbf{X}\mathbf{J}_\sigma^T\mathbf{V}_h\mathbf{W}_{\text{out},h}^T\mathbf{W}_{\text{out},h}\mathbf{V}_h^T\mathbf{J}_\sigma\mathbf{X}^T \otimes \frac{\mathbf{Q}_h^T\mathbf{Q}_h}{d_h}. \quad (11)$$

**Hessian for $\mathbf{W}_{V,h}$**  When quantizing $\mathbf{W}_{V,h}$, only $\mathbf{V}_h$ changes. Thus, the attention reconstruction error is expressed as

$$\|\Delta\text{MHA}(\mathbf{X})\|_F^2 = \|\mathbf{W}_{\text{out},h}(\mathbf{A}_h\Delta\mathbf{V}_h)^T\|_F^2 = \|\mathbf{W}_{\text{out},h}\Delta\mathbf{W}_{V,h}\mathbf{X}\mathbf{A}_h^T\|_F^2,$$

which yields the following Hessian for $\mathbf{W}_{V,h}$:

$$\mathbf{H}^{(\mathbf{w}_{V,h})} = 2\mathbf{X}\mathbf{A}_h^T\mathbf{A}_h\mathbf{X}^T \otimes \mathbf{W}_{\text{out},h}^T\mathbf{W}_{\text{out},h}. \quad (12)$$

Table 1: Proposed attention-aware Hessians

| Layer | $\mathbf{H}$ |
|-------|--------------|
| $\mathbf{W}_{Q,h}$ | $2\mathbf{X}\mathbf{X}^T \otimes \mathbf{K}_h^T\mathbf{K}_h$ |
| $\mathbf{W}_{K,h}$ | $2\mathbf{X}\mathbf{X}^T \otimes \mathbf{Q}_h^T\mathbf{Q}_h$ |
| $\mathbf{W}_{V,h}$ | $2\mathbf{X}\mathbf{A}_h^T\mathbf{A}_h\mathbf{X}^T \otimes \mathbf{W}_{\mathrm{out},h}^T\mathbf{W}_{\mathrm{out},h}$ |
| $\mathbf{W}_{\mathrm{out},h}$ | $2\mathbf{X}_{\mathrm{out},h}\mathbf{X}_{\mathrm{out},h}^T \otimes \mathbf{I}$ |
| $\mathbf{W}_{\mathrm{fc1}}$ | $2\mathbf{X}_{\mathrm{fc1}}\mathbf{X}_{\mathrm{fc1}}^T \otimes \mathbf{I}$ |
| $\mathbf{W}_{\mathrm{fc2}}$ | $2\mathbf{X}_{\mathrm{fc2}}\mathbf{X}_{\mathrm{fc2}}^T \otimes \mathbf{I}$ |

**Hessian for $\mathbf{W}_{\mathbf{out},h}$**   When $\mathbf{W}_{\mathrm{out},h}$ is quantized, the attention reconstruction error is expressed as

$$\|\Delta \mathrm{MHA}(\mathbf{X})\|_F^2 = \|\Delta\mathbf{W}_{\mathrm{out},h}(\mathbf{A}_h\mathbf{V}_h)^T\|_F^2.$$

Thus, the corresponding Hessian is obtained as

$$\mathbf{H}^{(\mathbf{w}_{out,h})} = 2\mathbf{V}_h^T\mathbf{A}_h^T\mathbf{A}_h\mathbf{V}_h \otimes \mathbf{I} = 2\mathbf{X}_{\mathrm{out}}\mathbf{X}_{\mathrm{out}}^T \otimes \mathbf{I}, \tag{13}$$

where $\mathbf{X}_{\mathrm{out},h} = (\mathbf{A}_h\mathbf{V}_h)^T$ is the input to the out-projection layer.

## 3.3   EFFICIENT IMPLEMENTATION OF BOA

While inter-layer dependencies within the attention module can be considered by exploiting the proposed Hessians, they are significantly more complex than the conventional Hessian in (4), which may incur high computational costs. For example, computing the proposed Hessians in (8) and (11) would be more expensive than computing the conventional one in (4). In this subsection, we present techniques for mitigating the computational overheads incurred by the proposed attention-aware Hessians.

**Hessian relaxation**   The largest overhead related to the computation of the proposed Hessians is the Jacobian matrix $\mathbf{J}_\sigma$ in (8) and (11). For one input sequence, the shape of $\mathbf{J}_\sigma$ is $H \times L \times L \times L$, which requires a large amount of memory and high computational cost (more than 400 GB even for the OPT-125M model when $L = 2048$).

To mitigate such computational overhead, we establish a relaxed Hessian that does not require the computation of $\mathbf{J}_\sigma$. To this end, we build an upper bound for the attention reconstruction error in (7), which will be used as its surrogate:

$$\|\Delta \mathrm{MHA}(\mathbf{X})\|_F^2 \leq \left\|\frac{\mathbf{W}_{\mathrm{out},h}\mathbf{V}_h^T\mathbf{J}_\sigma}{\sqrt{d_h}}\right\|_F^2 \cdot \|\mathbf{K}_h\Delta\mathbf{W}_{Q,h}\mathbf{X}\|_F^2. \tag{14}$$

Moreover, we note that the term $\|\mathbf{W}_{\mathrm{out},h}\mathbf{V}_h^T\mathbf{J}_\sigma\|_F^2$ in (14) is constant and does not affect quantization.[5] Thus, we do not need to consider this term when computing the Hessian. In short, we use the term $\|\mathbf{K}_h\Delta\mathbf{W}_{Q,h}\mathbf{X}\|_F^2$ as a surrogate of the attention reconstruction error when deriving the Hessian for $\mathbf{W}_{Q,h}$, which results in the following relaxed Hessian:

$$\mathbf{H}^{(\mathbf{w}_{Q,h})} = 2\mathbf{X}\mathbf{X}^T \otimes \mathbf{K}_h^T\mathbf{K}_h. \tag{15}$$

Similarly, we can establish a relaxed Hessian for $\mathbf{W}_{K,h}$ as follows:

$$\mathbf{H}^{(\mathbf{w}_{K,h})} = 2\mathbf{X}\mathbf{X}^T \otimes \mathbf{Q}_h^T\mathbf{Q}_h. \tag{16}$$

In Table 1, we summarize the relaxed Hessians for each layer inside the Transformer block.

**Efficient computation of inverse Hessians**   Owing to the size of the proposed attention-aware Hessians being $dd_h \times dd_h$, the complexity of the computation of the inverse Hessian (see (3)) would be $\mathcal{O}(d^3d_h^3)$ in our approach. This is considerably more expensive than the complexity $\mathcal{O}(d^3)$ in GPTQ, where the inverse of only the column-wise Hessian $\mathbf{X}\mathbf{X}^T \in \mathbb{R}^{d \times d}$ (in (4)) is needed (Frantar et al., 2023).

---

[5]The weight-update $\boldsymbol{\delta}\mathbf{w}$ in (3) is not affected by the constant multiple of $\mathbf{H}$ because $[c\mathbf{U}]_{q,:}/[c\mathbf{U}]_{q,q} = [\mathbf{U}]_{q,:}/[\mathbf{U}]_{q,q}$ for any constant $c$.

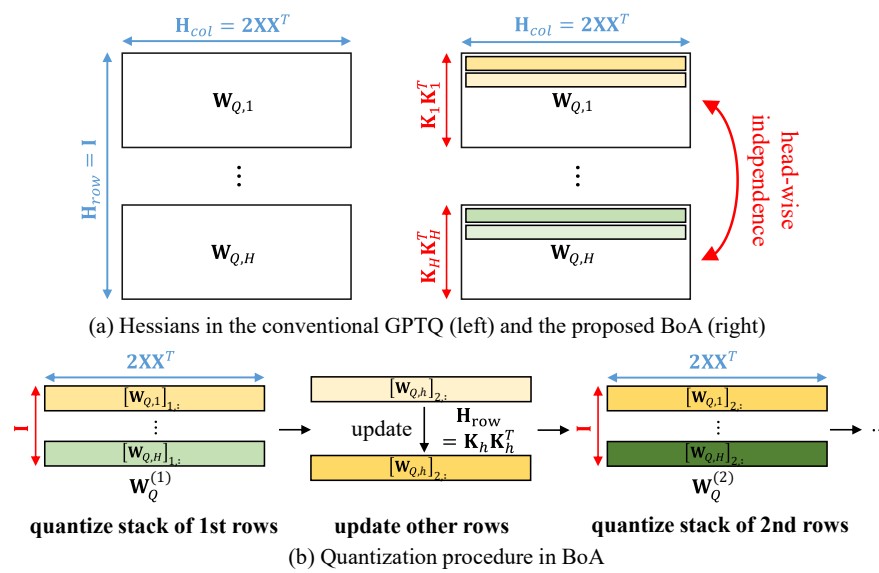

(a) Hessians in the conventional GPTQ (left) and the proposed BoA (right)

**quantize stack of 1st rows**    **update other rows**    **quantize stack of 2nd rows**

(b) Quantization procedure in BoA

Figure 1: Illustration of the proposed BoA for the query projection $\mathbf{W}_Q$.

For the efficient computation of the inverse Hessians, we exploit the useful properties of the Kronecker product (see (17a)-(17c) in Appendix A). For simplicity, let $\mathbf{H} = \mathbf{H}_{\text{col}} \otimes \mathbf{H}_{\text{row}}$ where $\mathbf{H}_{\text{col}} \in \mathbb{R}^{d \times d}$ and $\mathbf{H}_{\text{row}} \in \mathbb{R}^{d_h \times d_h}$, then we obtain

$$\mathbf{H}^{-1} = (\mathbf{H}_{\text{col}} \otimes \mathbf{H}_{\text{row}})^{-1} = \mathbf{H}_{\text{col}}^{-1} \otimes \mathbf{H}_{\text{row}}^{-1}.$$

This implies that the inverse Hessian $\mathbf{H}^{-1}$ can be computed by computing $\mathbf{H}_{\text{col}}^{-1}$ and $\mathbf{H}_{\text{row}}^{-1}$ (line 6 in Algorithm 1) whose complexity is $\mathcal{O}(d^3) + \mathcal{O}(d_h^3) (= \mathcal{O}(d^3))$, not $\mathcal{O}(d^3 d_h^3)$. Similarly, we can efficiently compute the Cholesky decomposition with the same order of complexity as in GPTQ. Specifically, if $\mathbf{L}_1 = \text{Chol}(\mathbf{H}_{\text{col}}^{-1})$ and $\mathbf{L}_2 = \text{Chol}(\mathbf{H}_{\text{row}}^{-1})$, $\mathbf{H}^{-1}$ can be expressed as

$$\mathbf{H}^{-1} = \mathbf{L}_1 \mathbf{L}_1^T \otimes \mathbf{L}_2 \mathbf{L}_2^T = (\mathbf{L}_1 \otimes \mathbf{L}_2)(\mathbf{L}_1 \otimes \mathbf{L}_2)^T.$$

Subsequently, noting that the Kronecker product of lower triangular matrices is also lower triangular, we obtain

$$\text{Chol}(\mathbf{H}^{-1}) = \mathbf{L}_1 \otimes \mathbf{L}_2 = \text{Chol}(\mathbf{H}_{\text{col}}^{-1}) \otimes \text{Chol}(\mathbf{H}_{\text{row}}^{-1}).$$

Thus, we can obtain $\text{Chol}(\mathbf{H}^{-1})$ by computing $\text{Chol}(\mathbf{H}_{\text{col}}^{-1})$ and $\text{Chol}(\mathbf{H}_{\text{row}}^{-1})$ (line 7 in Algorithm 1). Consequently, the computational complexity of the Choleksy decomposition would be $\mathcal{O}(d^3)$, not $\mathcal{O}(d^3 d_h^3)$.

**Simultaneous quantization of different heads**    The conventional Hessian in (4) implies the independence of different rows (see $\mathbf{H}_{\text{row}} = \mathbf{I}$). Whereas, the proposed attention-aware Hessians model the dependency between different rows (*e.g.*, $\mathbf{H}_{\text{row}} = \mathbf{K}_h^T \mathbf{K}_h$ for the query projection), using which we can compensate for the quantization error of a certain row by updating the other rows. However, in this case, the rows must be quantized sequentially (not simultaneously). For example, the second row can be quantized after being updated to compensate for the quantization error of the first row. This is in contrast to GPTQ where all the rows are quantized simultaneously.

To accelerate the quantization process, we assume independence between different attention heads (see Fig. 1(a)), under which rows related to different heads are independent and can thus be quantized together. For a better understanding, we consider the query projection $\mathbf{W}_Q$ as an example (see Fig. 1(b)). In the quantization step, we stack the $j$-th rows $[\mathbf{W}_{Q,h}]_{j,:}$ of all different heads, constructing the sub-weight matrix $\mathbf{W}_Q^{(j)} \in \mathbb{R}^{H \times d}$ (line 9 in Algorithm 1). Because the rows of $\mathbf{W}_Q^{(j)}$ are mutually independent, all the rows of $\mathbf{W}_Q^{(j)}$ can be quantized simultaneously as in GPTQ (line 10 in Algorithm 1). Following the quantization of $j$-th rows, we compensate for the quantization error by updating the remaining rows. In this update step, we use the refined weight-update formula (line 11 in Algorithm 1); the detailed derivation for this is provided in Appendix B.

To evaluate how much the quantization process can be accelerated by the head-wise simultaneous quantization, we measure the processing times of the proposed method with and without simultaneous quantization. From Table 2, we observe that BOA requires a significantly long processing time without the simultaneous quantization (more than one day for the 2.7B model). This is because all rows need to be quantized sequentially (*e.g.*, 2560 rows are quantized sequentially for OPT-2.7b) and thus the massive compute capabilities of modern GPUs cannot be utilized properly. As evident from Table 2, by applying the head-wise simultaneous quantization, we can achieve a significant reduction in the processing time.

Table 2: Processing time of BOA with and without head-wise simultaneous quantization

| Head-wise Simultaneous Quantization | Model Size (OPT) | | | |
|:---:|:---:|:---:|:---:|:---:|
| | 125M | 350M | 1.3B | 2.7B |
| X | 49.22 min | 181.7 min | 712.7 min | 24.48 hr |
| O | 5.099 min | 13.93 min | 31.64 min | 1.101 hr |

## 4 EXPERIMENTS

### 4.1 EXPERIMENTAL SETUP

To evaluate the performance of the proposed BOA, we quantize publicly available language models including OPT (Zhang et al., 2022), BLOOM (Scao et al., 2022), and LLaMA (Touvron et al., 2023)). As in (Frantar et al., 2023; Jeon et al., 2023b; Chee et al., 2023), we quantize only weights and retain activations with full precision because activations do not pose a significant bottleneck for the inference of LLMs (Frantar et al., 2023; Kim et al., 2023). As a calibration dataset, we use 128 random 2048 token segments from the C4 dataset (Raffel et al., 2020). Thus, we do not use any task-specific data for quantization. We evaluate the performance of the quantized models using benchmark datasets (*e.g.*, WikiText-2 (Merity et al., 2016), C4 (Raffel et al., 2020), and PTB (Marcus et al., 1993)) and zero-shot tasks. All experiments were conducted using a single NVIDIA A100 GPU (80 GB).

When determining a quantization order in BOA, the heuristic introduced by GPTQ can be employed; the column/row corresponding to the largest $\text{diag}(\mathbf{H}_{\text{col}})/\text{diag}(\mathbf{H}_{\text{row}})$ (*i.e.*, the most quantization-sensitive column/row) is first quantized for better compensation. Empirically, we observed that this heuristic could occasionally enhance the performance, yet at other times, it may result in inferior performance. We conduct experiments with and without this heuristic and report the better results.

### 4.2 COMPARISON WITH GPTQ

We compare the proposed BOA with GPTQ (Frantar et al., 2023), which is our primary baseline. For both algorithms, we set per-channel quantization parameters (*i.e.*, scale and zero-point) to minimize the layer-wise reconstruction error (line 5 in Algorithm 1). We note that in GPTQ, the Min-Max-based quantization parameters have been used (Frantar et al., 2023); however, this results in significantly worse quantization performance (Jeon et al., 2023b). While both algorithms aim to optimize the weight-rounding mechanism and can be combined with existing ET-based methods such as SmoothQuant (Xiao et al., 2023), AWQ (Lin et al., 2024), and Z-FOLD (Jeon et al., 2023b), we do not perform an equivalent transform in this experiment to solely compare the weight-rounding optimization performance. The results of integration with ET-based methods are presented in Section 4.3.

First, we compare the perplexity (PPL) performances of BOA and GPTQ (see Table 3 and Tables 7 and 8 in Appendix C.1). The performance of the rounding-to-nearest (RTN) method (which naively assigns the nearest quantized value) is also included for comparison, as in (Frantar et al., 2023). While RTN collapses for low bit-widths, BOA and GPTQ exhibit reasonable PPL, even for INT2 quantization. This is because BOA and GPTQ aim to minimize the task loss degradation, not the weight quantization error $\Delta \mathbf{W}$. Evidently, the proposed BOA outperforms GPTQ for all models. In particular, the performance gap is significant for low bit-width (*i.e.*, INT2) and small-sized models suited for resource-limited devices (*e.g.*, mobile devices).

Table 3: INT2 quantization performance (PPL ↓) of the proposed BOA and the conventional GPTQ.

(a) OPT

| Dataset | Method | 125M | 350M | 1.3B | 2.7B | 6.7B | 13B | 30B |
|---------|--------|------|------|------|------|------|-----|-----|
| WikiText-2 | RTN | 5.5e3 | 2.8e4 | 1.1e5 | 9.5e3 | 2.8e4 | 1.9e5 | 1.7e5 |
| | GPTQ | 232.8 | 98.65 | 66.76 | 37.44 | 24.74 | 18.97 | 13.12 |
| | **BoA** | **141.6** | **57.40** | **48.71** | **26.20** | **22.71** | **18.76** | **12.15** |
| PTB | RTN | 4.3e3 | 2.8e4 | 1.1e4 | 6.8e3 | 1.8e4 | 1.2e5 | 1.7e5 |
| | GPTQ | 384.8 | 135.9 | 112.0 | 64.59 | 42.36 | 26.95 | 20.25 |
| | **BoA** | **199.2** | **90.87** | **78.73** | **40.76** | **33.77** | **25.34** | **18.52** |
| C4 | RTN | 3.7e3 | 1.6e4 | 7.7e3 | 7.7e3 | 1.4e4 | 9.7e4 | 5.8e4 |
| | GPTQ | 178.6 | 71.89 | 64.11 | 33.94 | 24.86 | 20.08 | 14.45 |
| | **BoA** | **118.1** | **54.07** | **48.92** | **26.57** | **23.03** | **19.22** | **13.84** |

(b) BLOOM and LLaMA

| Dataset | Method | BLOOM | | | | | LLaMA | |
| | | 560M | 1.1B | 1.7B | 3B | 7.1B | 13B | 30B |
|---------|--------|------|------|------|------|------|-----|-----|
| WikiText-2 | RTN | 7.8e5 | 9.8e5 | 3.5e5 | 1.4e5 | 2.1e5 | 5.7e4 | 2.7e4 |
| | GPTQ | 59.23 | 43.93 | 36.48 | 29.25 | 20.20 | 12.67 | 8.844 |
| | **BoA** | **52.09** | **38.16** | **30.76** | **24.25** | **17.54** | **11.56** | **7.993** |
| PTB | RTN | 7.4e5 | 1.1e6 | 2.5e5 | 1.2e5 | 2.2e5 | 8.1e4 | 3.3e4 |
| | GPTQ | 142.6 | 176.4 | 95.32 | 67.48 | 43.73 | 20.55 | 14.64 |
| | **BoA** | **113.0** | **139.1** | **69.98** | **53.10** | **35.97** | **18.49** | **13.24** |
| C4 | RTN | 1.4e6 | 2.1e6 | 2.7e5 | 9.2e4 | 1.3e5 | 5.9e4 | 2.8e4 |
| | GPTQ | 57.31 | 43.48 | 38.69 | 30.97 | 23.52 | 14.24 | 11.78 |
| | **BoA** | **52.12** | **39.03** | **33.71** | **27.26** | **21.22** | **13.34** | **10.53** |

[*] INT3/INT4 quantization results are provided in Appendix C.1 due to the page limitation.

Table 4: INT2 zero-shot task performance (accuracy ↑) of the proposed BOA and GPTQ.

| Model | Size | Method | ARC-c | ARC-e | HellaSwag | MMLU | Average |
|-------|------|--------|-------|-------|-----------|------|---------|
| OPT | 1.3B | GPTQ | 22.53 | 35.61 | 34.03 | 22.93 | 28.78 |
| | | **BoA** | 22.53 | 38.72 | 36.00 | 23.12 | **30.09** |
| | 2.7B | GPTQ | 24.40 | 38.47 | 37.87 | 23.04 | 30.95 |
| | | **BoA** | 25.51 | 42.89 | 43.68 | 23.14 | **33.81** |
| | 6.7B | GPTQ | 25.60 | 42.85 | 43.29 | 24.09 | 33.96 |
| | | **BoA** | 26.62 | 44.91 | 44.52 | 24.33 | **35.10** |
| | 13B | GPTQ | 26.62 | 44.15 | 50.09 | 24.59 | 36.36 |
| | | **BoA** | 27.47 | 47.39 | 54.42 | 25.21 | **38.62** |
| | 30B | GPTQ | 31.57 | 52.99 | 60.55 | 25.27 | 42.60 |
| | | **BoA** | 31.48 | 53.24 | 62.58 | 26.41 | **43.43** |
| LLaMA | 13B | GPTQ | 32.17 | 58.71 | 57.48 | 23.53 | 42.97 |
| | | **BoA** | 33.79 | 59.01 | 59.73 | 23.90 | **44.11** |
| | 30B | GPTQ | 37.12 | 62.84 | 65.09 | 31.16 | 49.05 |
| | | **BoA** | 37.88 | 63.47 | 66.31 | 33.11 | **50.19** |

Next, we compare the zero-shot performances of BOA and GPTQ (see Table 4). To this end, we measure the accuracy of quantized models for several tasks and then average the results. We note that the zero-shot setting is maintained in our experiments because we do not use task-specific data for quantization. As evident, the proposed BOA outperforms GPTQ for all models. The key factor leading to such an outstanding performance is that we consider inter-layer dependencies within the attention module by targeting attention-wise reconstruction. This is in contrast to GPTQ, where layers are assumed to be independent.

Table 5: INT2 performance (PPL ↓) of BOA integrated with existing ET-based methods.

| Equivalent Transformation | Dataset | Method | Model Size (OPT) | | | | | | |
|---|---|---|---|---|---|---|---|---|---|
| | | | 125M | 350M | 1.3B | 2.7B | 6.7B | 13B | 30B |
| SmoothQuant | WikiText-2 | GPTQ | 229.4 | N/A | 39.88 | 27.31 | 20.03 | 15.32 | 13.55 |
| | | BOA | **151.2** | N/A | **31.62** | **24.45** | **18.55** | **14.29** | **12.49** |
| | PTB | GPTQ | 292.3 | N/A | 64.17 | 44.75 | 32.01 | 22.03 | 19.26 |
| | | BOA | **223.9** | N/A | **58.17** | **38.87** | **27.85** | **19.79** | **17.97** |
| | C4 | GPTQ | 151.4 | N/A | 38.13 | 26.80 | 21.22 | 16.19 | 14.42 |
| | | BOA | **130.4** | N/A | **34.20** | **24.95** | **20.92** | **15.33** | **13.90** |
| Z-FOLD | WikiText-2 | GPTQ | 156.0 | 102.5 | 33.97 | 27.10 | 18.07 | 16.29 | 13.24 |
| | | BOA | **107.9** | **54.72** | **29.38** | **23.96** | **17.18** | **15.14** | **12.41** |
| | PTB | GPTQ | 206.9 | 130.7 | 53.80 | 46.08 | 26.79 | 23.73 | 19.27 |
| | | BOA | **166.1** | **82.27** | **49.18** | **39.45** | **24.94** | **22.86** | **18.11** |
| | C4 | GPTQ | 108.8 | 71.37 | 31.67 | 25.98 | 19.79 | 17.21 | 14.13 |
| | | BOA | **86.07** | **49.39** | **28.65** | **24.19** | **19.01** | **16.17** | **13.67** |

[*] SmoothQuant does not support OPT-350M where the post-LayerNorm architecture has been used.

## 4.3 INTEGRATION WITH EQUIVALENT TRANSFORM-BASED METHODS

As mentioned, the performance of the proposed BOA can be enhanced by combining BOA with existing ET-based methods (*i.e.*, transforming models with ET-based methods first and then applying BOA for optimizing the weight-rounding mechanism). To verify this, we evaluate the performance of BOA integrated with ET-based methods. Among various algorithms, we use SmoothQuant (Xiao et al., 2023) and Z-FOLD (Jeon et al., 2023b) in our integration because they efficiently find out an equivalent transform without time-consuming gradient-based optimization.[6]

Table 5 and Table 9 (see Appendix C.2) summarize the PPL performances of the proposed BOA combined with SmoothQuant and Z-FOLD. For comparison, we also summarize the integration results for the conventional GPTQ. Overall, the performance of BOA indeed improves when combined with ET-based methods. We emphasize that the performance gap between the proposed BOA and GPTQ still remains significant for INT2 quantization. A similar behavior can be observed in the zero-shot results (see Table 10 in Appendix C.2); the performance is boosted by applying ET-based methods, and BOA outperforms GPTQ for all models regardless of the ET-based method.

## 4.4 COMPARISON WITH PRIOR ARTS

We compare the proposed BOA with OmniQuant (Shao et al., 2023) and AffineQuant (Ma et al., 2024), recently proposed algorithms that learn an attention-aware equivalent transform via back-propagation (see Table 6 and Table 11 in Appendix C.3 for PPL results and see Table 12 in Appendix C.3 for zero-shot results). In our comparison, we do not include AWQ (Lin et al., 2024) and OS+ (Wei et al., 2023) because they perform worse than OmniQuant and AffineQuant (Shao et al., 2023; Ma et al., 2024). Mixed quantization algorithms (*e.g.*, SpQR (Dettmers et al., 2023), SqueezeLLM (Kim et al., 2023), and OAC (Edalati et al., 2024)) and algorithms that require additional processing in the real inference stage (*e.g.*, QuIP (Chee et al., 2023)) are also not included because they require some dedicated kernels for acceleration which may not be supported by on-device NPUs such as Qualcomm Hexagon.

As evident, BOA itself outperforms existing algorithms in almost all cases, even though BOA does not rely on time-consuming gradient-based optimization and thus facilitates fast quantization (see Table 13 in Appendix C.4). Furthermore, when combined with SmoothQuant or Z-FOLD, the performance gap between BOA and OmniQuant/AffineQuant is significant, which demonstrates the efficacy of the proposed BOA. We observe that OmniQuant and AffineQuant sometimes diverge or collapse (*i.e.*, PPL is larger than $10^3$) for INT2 quantization. In fact, to supplement the INT2 quantization performance, group-wise quantization parameters have been additionally used in OmniQuant

---

[6]While SmoothQuant has been proposed in the context of weight-activation quantization, the smoothing factor ($\mathbf{s}_j = \max(|\mathbf{X}_j|)^\alpha / \max(|\mathbf{W}_j|)^{1-\alpha}$) used for the equivalent transformation can also be used for weight-only quantization by setting $\alpha = 0$.

Table 6: INT2 performance (PPL ↓) of BoA and existing approaches.

| Dataset | Method | 125M | 1.3B | 2.7B | 6.7B | 13B | 30B |
|---|---|---|---|---|---|---|---|
| | OmniQuant | NaN | NaN | NaN | 2.3e4 | 4.5e5 | 3.8e5 |
| | AffineQuant | 174.5 | NaN | 42.26 | 26.25 | 38.89 | 5.6e5 |
| WikiText-2 | **BoA** | 141.6 | 48.71 | 26.20 | 22.71 | 18.76 | **12.15** |
| | **BoA** + SmoothQuant | 151.2 | 31.62 | 24.45 | 18.55 | **14.29** | 12.49 |
| | **BoA** + Z-Fold | **107.9** | **29.38** | **23.96** | **17.18** | 15.14 | 12.41 |
| | OmniQuant | NaN | NaN | NaN | 5.0e4 | 3.7e5 | 2.9e5 |
| | AffineQuant | 254.2 | NaN | 55.58 | 37.36 | 50.10 | 3.1e5 |
| PTB | **BoA** | 199.2 | 78.73 | 40.76 | 33.77 | 25.34 | 18.52 |
| | **BoA** + SmoothQuant | 223.9 | 58.17 | **38.87** | 27.85 | **19.79** | **17.97** |
| | **BoA** + Z-Fold | **166.1** | **49.18** | 39.45 | **24.94** | 22.86 | 18.11 |
| | OmniQuant | NaN | NaN | NaN | 3.0e4 | 2.0e5 | 2.1e5 |
| | AffineQuant | 107.0 | NaN | 34.45 | 25.11 | 31.50 | 3.3e5 |
| C4 | **BoA** | 118.1 | 48.92 | 26.57 | 23.03 | 19.22 | 13.84 |
| | **BoA** + SmoothQuant | 130.4 | 34.20 | 24.95 | 20.92 | **15.33** | 13.90 |
| | **BoA** + Z-Fold | **86.07** | **28.65** | **24.19** | **19.01** | 16.17 | **13.67** |

\* 'NaN' means that loss diverges in the quantization process.

and AffineQuant, but group-wise parameters result in additional memory costs and processing time in the real inference step (Shen et al., 2023).

### 4.5 COMPARISON OF TIME AND MEMORY COSTS

We compare the processing time and memory costs of BoA and conventional algorithms (see Table 13 in Appendix C.4). We observe that the processing time of BoA is shorter than those required by existing attention-aware algorithms (*i.e.*, OmniQuant and AffineQuant), yet BoA achieves significantly better performance (see Tables 6 and 11), which demonstrates the efficacy of the proposed method. We also observe that BoA requires longer processing time and larger memory than those required by GPTQ. This is because GPTQ quantizes all the rows of the weight matrix simultaneously using only layer input. In contrast, BoA sequentially quantizes sub-weight matrices using outputs of other layers as well as layer input (see Fig. 1(b)) to consider the inter-layer dependencies within the attention module, which eventually leads to the better quantization performance than GPTQ.

Clearly, there is a trade-off between quantization speed / memory cost and accuracy. In real situations, when one needs to preserve the performance of the original model by considering inter-layer dependencies within the attention module, the proposed BoA would be an intriguing solution. Even when the memory resource is limited, BoA can be used with some relaxation. Specifically, we note that the large memory cost of BoA for hyper-scale LLMs (e.g., 13B and 30B) is attributable to the row-wise Hessian for the value projection ($\mathbf{X}\mathbf{A}_h^T\mathbf{A}_h\mathbf{X}^T$ in (12)) whose shape is $H \times d \times d$. In memory-limited cases, we can mitigate the memory cost of BoA by considering inter-layer dependencies only for query and key projections and applying the standard Hessian ($\mathbf{X}\mathbf{X}^T$ in (4)) for the value projection. Indeed, when applying the proposed Hessians only for query and key projections, BoA requires almost same amount of memory as GPTQ, yet still exhibiting better performance (see Table 14 in Appendix C.4). For more discussion on time and memory costs of the proposed BoA, see Appendix C.4.

### 5 CONCLUSION

In this paper, we proposed a novel PTQ algorithm called BoA. To consider the inter-layer dependencies within the attention module while circumventing time-consuming gradient-based optimization, we approximated the Hessian matrices by exploiting the attention reconstruction error. Furthermore, to mitigate the computational overhead incurred by the proposed attention-aware Hessians, we incorporated several techniques, such as Hessian relaxation, efficient computation of inverse Hessians, and head-wise simultaneous quantization. Finally, through extensive experiments, we demonstrated the efficacy of the proposed BoA algorithm.

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

## A    PROOF OF FOOTNOTE 4

In our proof, we use the following useful properties of the Kronecker product:

$$\text{vec}\left(\mathbf{M}_1\mathbf{M}_2\mathbf{M}_3\right) = \left(\mathbf{M}_3^T \otimes \mathbf{M}_1\right)\text{vec}(\mathbf{M}_2), \tag{17a}$$

$$\left(\mathbf{M}_1 \otimes \mathbf{M}_2\right)^T = \mathbf{M}_1^T \otimes \mathbf{M}_2^T, \tag{17b}$$

$$\left(\mathbf{M}_1 \otimes \mathbf{M}_2\right)\left(\mathbf{M}_3 \otimes \mathbf{M}_4\right) = \mathbf{M}_1\mathbf{M}_3 \otimes \mathbf{M}_2\mathbf{M}_4, \tag{17c}$$

where $\text{vec}(\cdot)$ denotes the vectorization operation.

Using (17a), we have

$$\|\mathbf{M}_1\Delta\mathbf{W}\mathbf{M}_2\|_F^2 = \left\|\left(\mathbf{M}_2^T \otimes \mathbf{M}_1\right)\Delta\mathbf{w}\right\|_2^2 = \Delta\mathbf{w}^T\left(\mathbf{M}_2^T \otimes \mathbf{M}_1\right)^T\left(\mathbf{M}_2^T \otimes \mathbf{M}_1\right)\Delta\mathbf{w},$$

where $\Delta\mathbf{w} = \text{vec}(\Delta\mathbf{W})$. In addition, by (17b) and (17c), we have

$$\Delta\mathbf{w}^T\left(\mathbf{M}_2^T \otimes \mathbf{M}_1\right)^T\left(\mathbf{M}_2^T \otimes \mathbf{M}_1\right)\Delta\mathbf{w} = \Delta\mathbf{w}^T\left(\mathbf{M}_2 \otimes \mathbf{M}_1^T\right)\left(\mathbf{M}_2^T \otimes \mathbf{M}_1\right)\Delta\mathbf{w}$$

$$= \Delta\mathbf{w}^T\left(\mathbf{M}_2\mathbf{M}_2^T \otimes \mathbf{M}_1^T\mathbf{M}_1\right)\Delta\mathbf{w}.$$

Finally, by exploiting the fact that $\frac{\partial^2 \mathbf{x}^T\mathbf{A}\mathbf{x}}{\partial\mathbf{x}^2} = \mathbf{A} + \mathbf{A}^T$, we obtain

$$\frac{\partial^2 \|\mathbf{M}_1\Delta\mathbf{W}\mathbf{M}_2\|_F^2}{\partial\Delta\mathbf{w}^2} = \mathbf{M}_2\mathbf{M}_2^T \otimes \mathbf{M}_1^T\mathbf{M}_1 + \left(\mathbf{M}_2\mathbf{M}_2^T \otimes \mathbf{M}_1^T\mathbf{M}_1\right)^T$$

$$\overset{(a)}{=} \mathbf{M}_2\mathbf{M}_2^T \otimes \mathbf{M}_1^T\mathbf{M}_1 + \left(\mathbf{M}_2\mathbf{M}_2^T\right)^T \otimes \left(\mathbf{M}_1^T\mathbf{M}_1\right)^T$$

$$= 2\mathbf{M}_2\mathbf{M}_2^T \otimes \mathbf{M}_1^T\mathbf{M}_1,$$

where (a) follows from (17b). This completes the proof.

## B    REFINED WEIGHT-UPDATE FORMULA

We recall that the Hessian-based weight-update formula is given by (Frantar & Alistarh, 2022; Frantar et al., 2023)

$$\boldsymbol{\delta w} = -\frac{w_q - \mathcal{Q}(w_q)}{[\mathbf{U}]_{q,q}}[\mathbf{U}]_{q,:} \text{ where } \mathbf{U} = \text{Chol}(\mathbf{H}^{-1})^T.$$

For the proposed attention-aware Hessians in Table 1, we have

$$\mathbf{U}_h = \mathbf{U}_{\text{col},h} \otimes \mathbf{U}_{\text{row},h},$$

where $\mathbf{U}_{\text{col},h} = \text{Chol}(\mathbf{H}_{\text{col},h}^{-1})^T$ and $\mathbf{U}_{\text{row},h} = \text{Chol}(\mathbf{H}_{\text{row},h}^{-1})^T$ (see Section 3.3). Therefore, the weight-update formula can be recast as

$$\boldsymbol{\delta w}_h = -\frac{w_q - \mathcal{Q}(w_q)}{[\mathbf{U}_{\text{col},h} \otimes \mathbf{U}_{\text{row},h}]_{q,q}}[\mathbf{U}_{\text{col},h} \otimes \mathbf{U}_{\text{row},h}]_{q,:}.$$

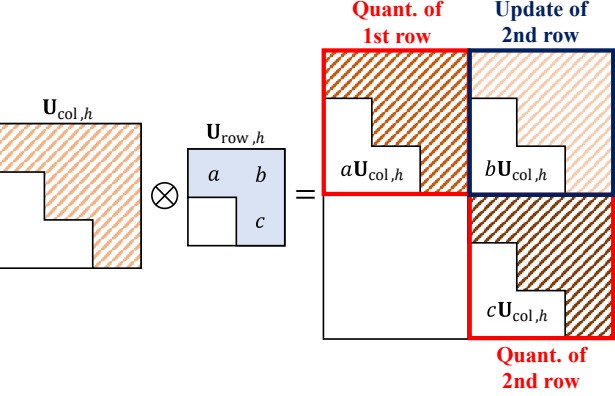

Figure 2: Illustration of the Hessian information when $d_{\text{row}} = 2$ and $d_{\text{col}} = 3$

For simplicity, suppose we quantize the first (0-th) row. When the weight $[\mathbf{W}_h]_{0,j}(= [\mathbf{W}^{(0)}]_{h,j})$ in the $j$-th column is quantized, the weight-update of the $i$-th row is simplified as (see Fig. 2 for the ease of understanding)

$$[\boldsymbol{\delta W}_h]_{i,:} = -\frac{[\mathbf{W}_h]_{0,j} - \mathcal{Q}([\mathbf{W}_h]_{0,j})}{[\mathbf{U}_{\text{row},h}]_{0,0}[\mathbf{U}_{\text{col},h}]_{j,j}}[\mathbf{U}_{\text{row},h}]_{0,i}[\mathbf{U}_{\text{col},h}]_{j,:}$$

$$= -\frac{[\mathbf{W}_h]_{0,j} - \mathcal{Q}([\mathbf{W}_h]_{0,j})}{[\mathbf{U}_{\text{col},h}]_{j,j}} \cdot \frac{[\mathbf{U}_{\text{row},h}]_{0,i}[\mathbf{U}_{\text{col},h}]_{j,:}}{[\mathbf{U}_{\text{row},h}]_{0,0}}$$

Thus, after the quantization of all weights in the first row, the total amount of the weight-update for the $i$-th row can be expressed as

$$[\boldsymbol{\delta W}_{h,\text{total}}]_{i,:} = -\sum_{j=0}^{d_{\text{col}}-1}\frac{[\mathbf{W}_h]_{0,j} - \mathcal{Q}([\mathbf{W}_h]_{0,j})}{[\mathbf{U}_{\text{col},h}]_{j,j}} \cdot \frac{[\mathbf{U}_{\text{row},h}]_{0,i}[\mathbf{U}_{\text{col},h}]_{j,:}}{[\mathbf{U}_{\text{row},h}]_{0,0}}$$

$$= -\frac{[\mathbf{U}_{\text{row},h}]_{0,i}}{[\mathbf{U}_{\text{row},h}]_{0,0}}\sum_{j=0}^{d_{\text{col}}-1}\frac{[\mathbf{W}_h]_{0,j} - \mathcal{Q}([\mathbf{W}_h]_{0,j})}{[\mathbf{U}_{\text{col},h}]_{j,j}} \cdot [\mathbf{U}_{\text{col},h}]_{j,:}.$$

Furthermore, by noting that (see line 8 in Algorithm 2)

$$[\mathbf{E}_{\text{GPTQ}}]_{h,j} = \frac{[\mathbf{W}_h]_{0,j} - \mathcal{Q}([\mathbf{W}_h]_{0,j})}{[\mathbf{U}_{\text{col},h}]_{j,j}},$$

we obtain

$$[\boldsymbol{\delta W}_{h,\text{total}}]_{i,:} = -\frac{[\mathbf{U}_{\text{row},h}]_{0,i}}{[\mathbf{U}_{\text{row},h}]_{0,0}}\sum_{j=0}^{d_{\text{col}}-1}[\mathbf{E}_{\text{GPTQ}}]_{h,j} \cdot [\mathbf{U}_{\text{col},h}]_{j,:} = -\frac{[\mathbf{U}_{\text{row},h}]_{0,i}}{[\mathbf{U}_{\text{row},h}]_{0,0}}[\mathbf{E}_{\text{GPTQ}}]_{h,:}\mathbf{U}_{\text{col},h}.$$

As a result, the weight-update matrix to compensate for the quantization error of the first row is given by

$$[\boldsymbol{\delta}\mathbf{W}_{h,\text{total}}]_{0:,:} = -\frac{[\mathbf{U}_{\text{row},h}^T]_{0:,0}[\mathbf{E}_{\text{GPTQ}}]_{h,:}\mathbf{U}_{\text{col},h}}{[\mathbf{U}_{\text{row},h}]_{0,0}}. \tag{18}$$

By taking similar steps as above, we can easily generalize (18) for the $j$-th row as follows:

$$[\boldsymbol{\delta}\mathbf{W}_{h,\text{total}}]_{j:,:} = -\frac{[\mathbf{U}_{\text{row},h}^T]_{j:,j}[\mathbf{E}_{\text{GPTQ}}]_{h,:}\mathbf{U}_{\text{col},h}}{[\mathbf{U}_{\text{row},h}]_{j,j}}. \tag{19}$$

## C    ADDITIONAL EXPERIMENTAL RESULTS

In this appendix, we provide experimental results omitted in the main text due to the page limitation.

### C.1    COMPARISON WITH GPTQ

Table 7 and Table 8 summarize the INT3/INT4 quantization performances (perplexity) of the proposed BOA and the conventional GPTQ on various sizes of OPT, BLOOM, and LLaMA models. As evident from Table 3, Table 7, and Table 8, BOA uniformly outperforms GPTQ, and the performance gap is significant for low bit-width (*i.e.*, INT2) and small-sized models suited for resource-limited devices (*e.g.*, mobile devices).

Table 7: Quantization performance (PPL ↓) of the proposed BOA and GPTQ on OPT.

(a) WikiText-2

| Precision | Method | 125M | 350M | 1.3B | 2.7B | 6.7B | 13B | 30B |
|---|---|---|---|---|---|---|---|---|
| FP16 | Baseline | 27.65 | 22.00 | 14.63 | 12.47 | 10.86 | 10.13 | 9.56 |
| INT4 | RTN | 37.28 | 25.94 | 48.20 | 16.92 | 12.10 | 11.32 | 10.98 |
| | GPTQ | 30.24 | 23.50 | 14.84 | 12.53 | 11.09 | 10.26 | 9.608 |
| | **BOA** | **28.93** | **22.90** | **14.72** | **12.44** | **10.88** | **10.16** | **9.571** |
| INT3 | RTN | 1.3e3 | 64.57 | 1.3e4 | 1.6e4 | 5.8e3 | 3.4e3 | 1.6e3 |
| | GPTQ | 38.74 | 26.31 | 16.70 | 14.01 | 11.91 | 10.85 | 9.911 |
| | **BOA** | **33.68** | **24.69** | **15.93** | **13.43** | **11.53** | **10.58** | **9.826** |

(b) PTB

| Precision | Method | 125M | 350M | 1.3B | 2.7B | 6.7B | 13B | 30B |
|---|---|---|---|---|---|---|---|---|
| FP16 | Baseline | 38.99 | 31.08 | 20.29 | 17.97 | 15.77 | 14.52 | 14.04 |
| INT4 | RTN | 53.88 | 36.79 | 75.37 | 32.41 | 18.86 | 16.41 | 15.44 |
| | GPTQ | 44.31 | 33.41 | 21.23 | 18.70 | 16.09 | 14.69 | **14.18** |
| | **BOA** | **41.50** | **32.58** | **21.02** | **18.42** | **15.90** | **14.63** | **14.18** |
| INT3 | RTN | 1.4e3 | 87.21 | 1.5e4 | 1.4e4 | 5.3e3 | 2.2e3 | 1.5e3 |
| | GPTQ | 57.62 | 39.35 | 24.77 | 21.53 | 17.56 | 15.68 | 14.56 |
| | **BOA** | **48.50** | **36.83** | **23.53** | **20.33** | **16.86** | **15.21** | **14.50** |

(c) C4

| Precision | Method | 125M | 350M | 1.3B | 2.7B | 6.7B | 13B | 30B |
|---|---|---|---|---|---|---|---|---|
| FP16 | Baseline | 26.56 | 22.59 | 16.07 | 14.34 | 12.71 | 12.06 | 11.44 |
| INT4 | RTN | 33.88 | 26.21 | 27.50 | 18.83 | 14.37 | 13.32 | 13.55 |
| | GPTQ | 28.53 | 23.73 | 16.51 | 14.72 | 12.88 | **12.16** | **11.50** |
| | **BOA** | **27.56** | **23.20** | **16.39** | **14.61** | **12.84** | **12.16** | 11.51 |
| INT3 | RTN | 834.4 | 55.15 | 6.6e3 | 1.2e4 | 5.0e3 | 2.8e3 | 1.8e3 |
| | GPTQ | 33.90 | 26.68 | 18.18 | 16.10 | 13.60 | 12.62 | 11.76 |
| | **BOA** | **31.12** | **25.39** | **17.74** | **15.83** | **13.34** | **12.52** | **11.75** |

Table 8: Quantization performance (PPL ↓) of BoA and GPTQ on BLOOM and LLaMA.

(a) WikiText-2

| Precision | Method | BLOOM | | | | | LLaMA | |
|---|---|---|---|---|---|---|---|---|
| | | 560M | 1.1B | 1.7B | 3B | 7.1B | 13B | 30B |
| FP16 | Baseline | 22.42 | 17.69 | 15.39 | 13.48 | 11.37 | 5.091 | 4.101 |
| INT4 | RTN | 25.82 | 19.98 | 16.96 | 14.75 | 12.09 | 5.525 | 4.536 |
| | GPTQ | 23.44 | 18.54 | 15.90 | 13.90 | 11.63 | 5.262 | 4.285 |
| | **BoA** | **23.28** | **18.32** | **15.81** | **13.84** | **11.58** | **5.243** | **4.262** |
| INT3 | RTN | 56.74 | 49.85 | 63.37 | 39.07 | 17.35 | 11.78 | 14.87 |
| | GPTQ | 26.63 | 20.80 | 17.71 | 15.39 | 12.42 | 5.721 | 4.848 |
| | **BoA** | **25.90** | **20.28** | **17.12** | **14.91** | **12.19** | **5.676** | **4.725** |

(b) PTB

| Precision | Method | BLOOM | | | | | LLaMA | |
|---|---|---|---|---|---|---|---|---|
| | | 560M | 1.1B | 1.7B | 3B | 7.1B | 13B | 30B |
| FP16 | Baseline | 43.69 | 57.96 | 30.00 | 25.34 | 20.83 | 9.081 | 8.159 |
| INT4 | RTN | 50.96 | 66.79 | 33.52 | 27.65 | 22.40 | 9.775 | 8.653 |
| | GPTQ | 45.33 | 61.94 | 31.37 | 26.39 | 21.40 | 9.306 | 8.344 |
| | **BoA** | **44.92** | **61.40** | **30.67** | **26.23** | **21.34** | **9.255** | **8.304** |
| INT3 | RTN | 124.8 | 184.0 | 105.5 | 66.24 | 34.94 | 28.94 | 28.79 |
| | GPTQ | 52.39 | 70.68 | 35.06 | 28.99 | 23.46 | 9.928 | 8.925 |
| | **BoA** | **50.71** | **67.77** | **33.92** | **28.67** | **22.86** | **9.857** | **8.737** |

(c) C4

| Precision | Method | BLOOM | | | | | LLaMA | |
|---|---|---|---|---|---|---|---|---|
| | | 560M | 1.1B | 1.7B | 3B | 7.1B | 13B | 30B |
| FP16 | Baseline | 26.60 | 22.05 | 19.49 | 17.49 | 15.20 | 6.798 | 6.131 |
| INT4 | RTN | 29.80 | 24.42 | 21.24 | 18.75 | 16.05 | 7.232 | 6.537 |
| | GPTQ | 27.39 | 22.69 | 20.03 | 17.89 | 15.44 | 6.973 | 6.294 |
| | **BoA** | **27.23** | **22.54** | **19.90** | **17.82** | **15.42** | **6.958** | **6.267** |
| INT3 | RTN | 66.99 | 60.41 | 113.6 | 79.84 | 22.54 | 14.46 | 30.04 |
| | GPTQ | 29.89 | 24.48 | 21.44 | 19.07 | 16.24 | 7.504 | 6.840 |
| | **BoA** | **29.39** | **24.17** | **21.02** | **18.74** | **16.09** | **7.454** | **6.718** |

## C.2 INTEGRATION WITH EQUIVALENT TRANSFORM-BASED METHODS

In this appendix, we verify that the performance of the proposed BOA can be enhanced by combining BOA with existing ET-based methods. Among various algorithms, we use SmoothQuant (Xiao et al., 2023) and Z-FOLD (Jeon et al., 2023b) in our integration because they efficiently find out an equivalent transform without time-consuming gradient-based optimization. We note that while SmoothQuant has been proposed in the context of weight-activation quantization, the smoothing factor ($\mathbf{s}_j = \max(|\mathbf{X}_j|)^{\alpha} / \max(|\mathbf{W}_j|)^{1-\alpha}$) used for the equivalent transformation can also be used for weight-only quantization by setting $\alpha = 0$.

Tables 5 and 9 summarize the integration results. Overall, the performances of BOA and GPTQ indeed improve when combined with ET-based methods. We emphasize that the performance gap between the proposed BOA and GPTQ still remains significant, especially for INT2 quantization. A similar behavior can be observed in the zero-shot results (see Table 10); the performance is boosted by applying ET-based methods, and BOA outperforms GPTQ regardless of the ET-based method.

Table 9: INT3 performance (PPL ↓) of BOA integrated with existing ET-based methods.

| Equivalent Transformation | Dataset | Method | Model Size (OPT) | | | | | | |
|---|---|---|---|---|---|---|---|---|---|
| | | | 125M | 350M | 1.3B | 2.7B | 6.7B | 13B | 30B |
| SmoothQuant | Wiki2 | GPTQ | 39.56 | N/A | 16.32 | 13.55 | 11.90 | 10.68 | 9.857 |
| | | **BOA** | **34.58** | N/A | **15.83** | **13.33** | **11.58** | **10.38** | **9.846** |
| | PTB | GPTQ | 58.00 | N/A | 24.00 | 20.36 | 17.18 | 15.45 | 14.46 |
| | | **BOA** | **51.44** | N/A | **22.78** | **19.83** | **16.74** | **15.18** | **14.40** |
| | C4 | GPTQ | 34.98 | N/A | 17.78 | 15.68 | 13.50 | 12.55 | 11.74 |
| | | **BOA** | **31.52** | N/A | **17.43** | **15.48** | **13.35** | **12.48** | **11.73** |
| Z-FOLD | Wiki2 | GPTQ | 39.59 | 25.97 | 16.10 | 13.54 | 11.65 | 10.64 | 9.887 |
| | | **BOA** | **33.31** | **24.22** | **15.91** | **13.33** | **11.28** | **10.53** | **9.814** |
| | PTB | GPTQ | 53.08 | 39.23 | 22.73 | 20.18 | 16.64 | 15.22 | 14.57 |
| | | **BOA** | **46.59** | **36.80** | **22.29** | **19.54** | **16.44** | **15.16** | **14.53** |
| | C4 | GPTQ | 33.67 | 26.45 | 17.33 | 15.50 | 13.28 | 12.46 | 11.73 |
| | | **BOA** | **30.00** | **25.04** | **17.13** | **15.32** | **13.20** | **12.41** | **11.71** |

[*] SmoothQuant does not support OPT-350M where the post-LayerNorm architecture has been used.

Table 10: INT2 zero-shot performance (accuracy ↑) of BOA integrated with ET-based methods.

| Equivalent Transformation | Model Size (OPT) | Method | Tasks | | | | Average |
|---|---|---|---|---|---|---|---|
| | | | ARC-c | ARC-e | HellaSwag | MMLU | |
| SmoothQuant | 1.3B | GPTQ | 23.38 | 40.15 | 37.47 | 23.00 | 31.00 |
| | | **BOA** | 22.18 | 42.00 | 37.48 | 23.10 | **31.19** |
| | 2.7B | GPTQ | 25.85 | 42.72 | 42.46 | 23.10 | 33.53 |
| | | **BOA** | 27.73 | 44.70 | 44.24 | 22.95 | **34.91** |
| | 6.7B | GPTQ | 25.68 | 46.25 | 45.24 | 23.40 | 35.14 |
| | | **BOA** | 27.56 | 48.15 | 46.20 | 23.90 | **36.45** |
| | 13B | GPTQ | 29.52 | 52.53 | 56.63 | 24.90 | 40.90 |
| | | **BOA** | 31.57 | 53.66 | 58.69 | 25.35 | **42.32** |
| | 30B | GPTQ | 29.18 | 54.46 | 60.04 | 24.54 | 42.06 |
| | | **BOA** | 31.57 | 55.93 | 62.18 | 25.54 | **43.81** |
| Z-FOLD | 1.3B | GPTQ | 23.89 | 42.05 | 40.45 | 23.07 | 32.37 |
| | | **BOA** | 24.74 | 43.98 | 40.31 | 23.45 | **33.12** |
| | 2.7B | GPTQ | 25.00 | 41.46 | 43.13 | 23.16 | 33.19 |
| | | **BOA** | 26.37 | 43.06 | 45.18 | 23.50 | **34.53** |
| | 6.7B | GPTQ | 30.46 | 48.78 | 52.46 | 25.64 | 39.34 |
| | | **BOA** | 28.58 | 49.75 | 55.45 | 26.67 | **40.11** |
| | 13B | GPTQ | 28.58 | 48.78 | 55.38 | 24.75 | 39.37 |
| | | **BOA** | 28.84 | 49.87 | 58.32 | 24.60 | **40.41** |
| | 30B | GPTQ | 31.83 | 53.70 | 61.34 | 24.96 | 42.96 |
| | | **BOA** | 30.12 | 57.53 | 63.63 | 24.85 | **44.03** |

## C.3 COMPARISON WITH PRIOR ARTS

We compare the proposed BOA with OmniQuant (Shao et al., 2023) and AffineQuant (Ma et al., 2024), recently proposed algorithms that learn an attention-aware equivalent transform via back-propagation (see Tables 6 and 11 for PPL results and see Table 12 for zero-shot results).

As evident, BOA itself outperforms existing algorithms in almost all cases, even though BOA does not rely on time-consuming gradient-based optimization. Furthermore, when combined with SmoothQuant or Z-FOLD, the performance gap between BOA and OmniQuant/AffineQuant is significant, which demonstrates the efficacy of BOA. We note that OmniQuant and AffineQuant sometimes diverge or collapse (*i.e.*, PPL is larger than $10^3$) for INT2 quantization. In fact, to supplement the INT2 quantization performance, group-wise quantization parameters have been additionally used in OmniQuant and AffineQuant, but group-wise parameters result in additional memory costs and processing time for the inference (Shen et al., 2023).

Table 11: INT3 performance (PPL ↓) of BOA and existing attention-aware approaches.

| Dataset | Method | Model Size (OPT) | | | | | |
|---|---|---|---|---|---|---|---|
| | | 125M | 1.3B | 2.7B | 6.7B | 13B | 30B |
| Wiki2 | OmniQuant | 41.59 | 18.23 | 15.11 | 12.86 | 12.49 | 11.26 |
| | AffineQuant | 37.75 | 17.12 | 14.32 | 12.42 | 11.93 | 10.72 |
| | **BOA** | 33.68 | 15.93 | 13.43 | 11.53 | 10.58 | 9.826 |
| | **BOA** + SmoothQuant | 34.58 | **15.83** | **13.33** | 11.58 | **10.38** | 9.846 |
| | **BOA** + Z-FOLD | **33.31** | 15.91 | **13.33** | **11.28** | 10.53 | **9.814** |
| PTB | OmniQuant | 59.51 | 26.08 | 22.68 | 18.31 | 17.76 | 16.02 |
| | AffineQuant | 53.02 | 24.47 | 21.18 | 17.27 | 17.27 | 15.31 |
| | **BOA** | 48.50 | 23.53 | 20.33 | 16.86 | 15.21 | 14.50 |
| | **BOA** + SmoothQuant | 51.44 | 22.78 | 19.83 | 16.74 | 15.18 | **14.40** |
| | **BOA** + Z-FOLD | **46.59** | **22.29** | **19.54** | **16.44** | **15.16** | 14.53 |
| C4 | OmniQuant | 35.73 | 19.10 | 16.80 | 14.40 | 13.48 | 12.44 |
| | AffineQuant | 33.37 | 18.56 | 16.15 | 13.91 | 13.31 | 12.14 |
| | **BOA** | 31.12 | 17.74 | 15.83 | 13.34 | 12.52 | 11.75 |
| | **BOA** + SmoothQuant | 31.52 | 17.43 | 15.48 | 13.35 | 12.48 | 11.73 |
| | **BOA** + Z-FOLD | **30.00** | **17.13** | **15.32** | **13.20** | **12.41** | **11.71** |

Table 12: Zero-shot task performance (accuracy↑) of BOA and existing attention-aware methods

| Model Size (OPT) | Method | Tasks | | | | Average |
|---|---|---|---|---|---|---|
| | | ARC-c | ARC-e | HellaSwag | MMLU | |
| 1.3B | OmniQuant | NaN | NaN | NaN | NaN | NaN |
| | AffineQuant | NaN | NaN | NaN | NaN | NaN |
| | **BOA** + SmoothQuant | 22.18 | 42.00 | 37.48 | 23.10 | 31.19 |
| | **BOA** + Z-FOLD | 24.74 | 43.98 | 40.31 | 23.45 | **33.12** |
| 2.7B | OmniQuant | NaN | NaN | NaN | NaN | NaN |
| | AffineQuant | 25.09 | 40.66 | 39.35 | 22.90 | 32.00 |
| | **BOA** + SmoothQuant | 27.73 | 44.70 | 44.24 | 22.95 | **34.91** |
| | **BOA** + Z-FOLD | 26.37 | 43.06 | 45.18 | 23.50 | 34.53 |
| 6.7B | OmniQuant | 23.55 | 28.87 | 25.60 | 22.95 | 25.24 |
| | AffineQuant | 26.62 | 48.23 | 47.18 | 23.57 | 36.40 |
| | **BOA** + SmoothQuant | 27.56 | 48.15 | 46.20 | 23.90 | 36.45 |
| | **BOA** + Z-FOLD | 28.58 | 49.75 | 55.45 | 26.67 | **40.11** |
| 13B | OmniQuant | 26.11 | 26.35 | 25.54 | 22.95 | 25.24 |
| | AffineQuant | 24.15 | 43.48 | 45.62 | 22.89 | 34.04 |
| | **BOA** + SmoothQuant | 31.57 | 53.66 | 58.69 | 25.35 | **42.32** |
| | **BOA** + Z-FOLD | 28.84 | 49.87 | 58.32 | 24.60 | 40.41 |
| 30B | OmniQuant | 26.11 | 26.73 | 25.83 | 22.95 | 25.41 |
| | AffineQuant | 25.26 | 26.47 | 25.65 | 22.95 | 25.08 |
| | **BOA** + SmoothQuant | 31.57 | 55.93 | 62.18 | 25.54 | 43.81 |
| | **BOA** + Z-FOLD | 30.12 | 57.53 | 63.63 | 24.85 | **44.03** |

[*] 'NaN' means that loss diverges in the quantization process.

## C.4 COMPARISON OF TIME AND MEMORY COSTS

In this appendix, we compare the processing time and memory costs of BOA and conventional algorithms.

Table 13: Time and memory costs of the proposed BOA and existing methods

(a) INT2 quantization processing time

| Method | Reconstruction Target | Model Size (OPT) | | | | | |
|---|---|---|---|---|---|---|---|
| | | 125M | 1.3B | 2.7B | 6.7B | 13B | 30B |
| GPTQ | Layer output | 0.752 min | 6.284 min | 0.214 hr | 0.603 hr | 1.293 hr | 3.689 hr |
| OmniQuant | Transformer block output | 16.20 min | 61.20 min | 1.627 hr | 2.933 hr | 5.309 hr | 11.57 hr |
| AffineQuant | Transformer block output | 28.33 min | 154.2 min | 4.597 hr | 9.854 hr | 18.41 hr | 44.25 hr |
| **BOA** | Attention output | 5.099 min | 31.64 min | 1.101 hr | 2.830 hr | 4.964 hr | 10.55 hr |

(b) Memory cost (GB)

| Method | Reconstruction Target | Model Size (OPT) | | | | | |
|---|---|---|---|---|---|---|---|
| | | 125M | 1.3B | 2.7B | 6.7B | 13B | 30B |
| GPTQ | Layer output | 1.391 | 3.970 | 4.863 | 9.011 | 12.57 | 21.25 |
| OmniQuant | Transformer block output | 1.941 | 5.869 | 7.095 | 11.68 | 16.15 | 26.94 |
| AffineQuant | Transformer block output | 3.473 | 9.963 | 12.25 | 20.08 | 26.78 | 42.21 |
| **BOA** | Attention output | 1.676 | 5.471 | 6.837 | 12.26 | 19.06 | 39.74 |

**Discussion on BOA**    The main reason why the processing time of the proposed BOA increases with the size of LLMs is that the embedding dimension of attention heads increases with the model size. Specifically, to compensate for the error incurred by the quantization of certain rows, BOA needs to sequentially quantize sub-weight matrix $d_h$ times where $d_h$ is the head dimension (see Fig. 1(b)). Because $d_h$ increases with model size, the number of sequential quantizations also increases, which leads to long processing time.

**Comparison with OmniQuant/AffineQuant**    The processing time of the proposed BOA is shorter than those required by existing attention-aware algorithms that rely on gradient-based optimization, yet achieving significantly better performance (see Tables 6 and 11). We note that OmniQuant does not take too much time for quantizing large LLMs even though it performs gradient-based optimization. This is because OmniQuant reduces the number of learnable parameters greatly to accelerate gradient-based optimization. Specifically, instead of learning a weight-rounding policy which requires to learn a large number of parameters, OmniQuant learns only a small number of quantization parameters and some parameters related to the equivalent transform (Shao et al., 2023). While this strategy accelerates the quantization process greatly, OmniQuant suffers from unstable quantization process or collapses for low-bit quantization (see Tables 6 and 11). AffineQuant improves OmniQuant by introducing additional learnable parameters (Ma et al., 2024), but such additional parameters result in huge processing time (4 times longer processing time; see Table 13(a)), which demonstrates the inefficiency of gradient-based optimization over the proposed method.

**Comparison with GPTQ**    The proposed BOA requires longer processing time and larger memory than those required by GPTQ. This is because GPTQ quantizes all the rows of the weight matrix simultaneously using only layer input. In contrast, BOA sequentially quantizes sub-weight matrices using outputs of other layers as well as layer input (see Fig. 1(b)) to consider the inter-layer dependencies within the attention module, which eventually leads to the better quantization performance than GPTQ. Clearly, there is a trade-off between quantization speed / memory cost and accuracy. In real situations, when one needs to preserve the performance of the original model by considering inter-layer dependencies within the attention module, the proposed BOA would be an intriguing solution. Even when the memory resource is limited, BOA can be used with some relaxation. Specifically, we note that the large memory cost of BOA for hyper-scale LLMs (e.g., 13B and 30B) is attributable to the row-wise Hessian for the value projection ($\mathbf{X}\mathbf{A}_h^T\mathbf{A}_h\mathbf{X}^T$ in (12)) whose shape is $H \times d \times d$ ($H$ is the number of attention heads and $d$ is the embedding dimension). In

memory-limited cases, we can mitigate the memory cost of BOA by considering inter-layer dependencies only for query and key projections and applying the standard Hessian ($\mathbf{X}\mathbf{X}^T$ in (4)) for the value projection. Indeed, when considering only query and key projections, BOA requires almost same amount of memory as GPTQ, yet still exhibiting better performance (see Table 14).

Table 14: Performance and memory costs of the proposed BOA with and without considering inter-layer dependencies for the value projection

(a) INT2 performance (PPL ↓) on C4

| Method | Consideration of Inter-layer Dependencies | Model Size (OPT) | | | | | |
|---|---|---|---|---|---|---|---|
| | | 125M | 1.3B | 2.7B | 6.7B | 13B | 30B |
| GPTQ | - | 178.6 | 64.11 | 33.94 | 24.86 | 20.08 | 14.45 |
| OmniQuant | query, key, value, out, fcs | NaN | NaN | NaN | 3.0e4 | 2.0e5 | 2.1e5 |
| AffineQuant | query, key, value, out, fcs | 107.0 | NaN | 34.45 | 25.11 | 31.50 | 3.3e5 |
| **BOA** | query, key | 130.8 | 52.84 | 27.81 | 23.95 | 19.98 | 14.11 |
| **BOA** | query, key, value | 118.1 | 48.92 | 26.57 | 23.03 | 19.22 | 13.84 |

(b) Memory cost (GB)

| Method | Consideration of Inter-layer Dependencies | Model Size (OPT) | | | | | |
|---|---|---|---|---|---|---|---|
| | | 125M | 1.3B | 2.7B | 6.7B | 13B | 30B |
| GPTQ | - | 1.391 | 3.970 | 4.863 | 9.011 | 12.57 | 21.25 |
| OmniQuant | query, key, value, out, fcs | 1.941 | 5.869 | 7.095 | 11.68 | 16.15 | 26.94 |
| AffineQuant | query, key, value, out, fcs | 3.473 | 9.963 | 12.25 | 20.08 | 26.78 | 42.21 |
| **BOA** | query, key | 1.391 | 3.971 | 4.864 | 9.015 | 12.57 | 21.25 |
| **BOA** | query, key, value | 1.676 | 5.471 | 6.837 | 12.26 | 19.06 | 39.74 |

* The additional memory required for the query and key projections, i.e., memory needed to save $\mathbf{K}_h^T\mathbf{K}_h$ and $\mathbf{Q}_h^T\mathbf{Q}_h$ (see (15) and (16)), is negligible (e.g., 0.003 GB for 30B).

## C.5 RESULTS FOR DIFFERENT CALIBRATION DATASETS

When constructing a calibration dataset, we randomly sample 128 sequences from the C4 dataset (see Section 4.1). By changing the seed for the sampling, we can obtain different calibration datasets, which leads to different quantization results.[7] In this appendix, we report the corresponding results and overall statistics. Due to the limited computational resources, we conducted this experiment only for our main comparison (*i.e.*, the performances of the proposed BOA and the conventional GPTQ).

Table 15: Performance (perplexity ↓) of the proposed BOA and GPTQ for different seeds.

(a) INT2 Quantization

| Dataset | Seed | Method | 125M | 1.3B | 2.7B | 6.7B | 13B | 30B |
|---------|------|--------|------|------|------|------|-----|-----|
| Wiki2 | 0 | GPTQ | 232.8 | 66.76 | 37.44 | 24.74 | 18.97 | 13.12 |
| | | BOA | **141.6** | **48.71** | **26.20** | **22.71** | **18.76** | **12.15** |
| | 10 | GPTQ | 276.2 | 66.30 | 36.74 | 24.64 | 20.05 | 13.34 |
| | | BOA | **147.4** | **47.23** | **25.95** | **23.11** | **18.52** | **12.17** |
| | 20 | GPTQ | 243.9 | 65.09 | 36.33 | 24.94 | 19.78 | 13.17 |
| | | BOA | **139.4** | **45.11** | **27.00** | **24.06** | **17.82** | **12.33** |
| | 50 | GPTQ | 269.9 | 64.88 | 33.84 | 24.54 | 19.47 | 13.41 |
| | | BOA | **160.7** | **44.13** | **26.75** | **23.09** | **18.70** | **12.15** |
| | 100 | GPTQ | 228.3 | 71.51 | 36.72 | 25.55 | 19.39 | 13.18 |
| | | BOA | **147.8** | **47.43** | **26.85** | **23.61** | **18.49** | **12.17** |
| | Mean | GPTQ | 250.2±22 | 66.91±2.7 | 36.21±1.4 | 24.88±0.40 | 19.53±0.41 | 13.24±0.12 |
| | ±Stdev | BOA | **147.4±8.3** | **46.52±1.9** | **26.55±0.45** | **23.32±0.53** | **18.46±0.38** | **12.19±0.077** |
| PTB | 0 | GPTQ | 384.8 | 112.0 | 64.59 | 42.36 | 26.95 | 20.25 |
| | | BOA | **199.2** | **78.73** | **40.76** | **33.77** | **25.34** | **18.52** |
| | 10 | GPTQ | 324.1 | 112.7 | 62.42 | 38.91 | 27.92 | 19.80 |
| | | BOA | **185.4** | **76.02** | **39.73** | **33.79** | **23.55** | **18.08** |
| | 20 | GPTQ | 350.4 | 111.6 | 62.64 | 39.84 | 27.80 | 20.57 |
| | | BOA | **188.7** | **79.31** | **41.69** | **34.89** | **24.59** | **18.06** |
| | 50 | GPTQ | 433.8 | 122.1 | 59.46 | 43.05 | 27.64 | 20.43 |
| | | BOA | **206.8** | **87.29** | **41.91** | **36.36** | **24.56** | **18.27** |
| | 100 | GPTQ | 479.2 | 125.8 | 59.49 | 38.26 | 27.56 | 20.02 |
| | | BOA | **164.8** | **77.32** | **41.67** | **35.19** | **24.56** | **17.98** |
| | Mean | GPTQ | 394.5±63 | 116.8±6.6 | 61.72±2.2 | 40.48±2.1 | 27.57±0.38 | 20.21±0.31 |
| | ±Stdev | BOA | **189.0±16** | **79.73±4.4** | **41.15±0.91** | **34.80±1.1** | **24.52±0.64** | **18.18±0.22** |
| C4 | 0 | GPTQ | 178.6 | 64.11 | 33.94 | 24.86 | 20.08 | 14.45 |
| | | BOA | **118.1** | **48.92** | **26.57** | **23.03** | **19.22** | **13.84** |
| | 10 | GPTQ | 189.7 | 64.19 | 33.27 | 24.40 | 20.40 | 14.44 |
| | | BOA | **115.8** | **49.76** | **27.00** | **24.04** | **18.58** | **13.83** |
| | 20 | GPTQ | 163.1 | 64.19 | 32.83 | 24.66 | 20.37 | 14.41 |
| | | BOA | **111.1** | **48.81** | **26.96** | **23.31** | **19.15** | **13.80** |
| | 50 | GPTQ | 190.8 | 64.99 | 32.69 | 24.55 | 20.13 | 14.48 |
| | | BOA | **119.2** | **46.68** | **27.11** | **23.73** | **18.96** | **13.80** |
| | 100 | GPTQ | 168.5 | 67.54 | 33.65 | 25.25 | 20.23 | 14.51 |
| | | BOA | **116.7** | **48.49** | **27.17** | **24.09** | **19.63** | **13.91** |
| | Mean | GPTQ | 178.2±12 | 65.00±1.5 | 33.28±0.53 | 24.75±0.33 | 20.24±0.14 | 14.46±0.039 |
| | ±Stdev | BOA | **116.2±3.1** | **48.53±1.1** | **26.96±0.24** | **23.64±0.46** | **19.11±0.38** | **13.83±0.045** |

---

[7]Tables 3 to 11 present the results for seed 0.

(b) INT3 Quantization

| Dataset | Seed | Method | 125M | 1.3B | 2.7B | 6.7B | 13B | 30B |
|---|---|---|---|---|---|---|---|---|
| Wiki2 | 0 | GPTQ | 38.74 | 16.70 | 14.01 | 11.91 | 10.85 | 9.911 |
| | | **BoA** | **33.68** | **15.93** | **13.43** | **11.53** | **10.58** | **9.826** |
| | 10 | GPTQ | 40.72 | 16.99 | 13.97 | 11.89 | 10.84 | 9.881 |
| | | **BoA** | **34.30** | **16.15** | **13.43** | **11.43** | **10.59** | **9.756** |
| | 20 | GPTQ | 38.72 | 17.17 | 13.89 | 11.98 | 10.94 | 9.941 |
| | | **BoA** | **34.24** | **16.02** | **13.32** | **11.66** | **10.51** | **9.791** |
| | 50 | GPTQ | 37.72 | 16.93 | 14.08 | 11.86 | 10.92 | 9.769 |
| | | **BoA** | **34.73** | **16.19** | **13.44** | **11.44** | **10.54** | **9.768** |
| | 100 | GPTQ | 38.37 | 16.97 | 14.27 | 11.89 | 10.88 | **9.840** |
| | | **BoA** | **34.51** | **16.14** | **13.72** | **11.49** | **10.52** | 9.847 |
| | Mean | GPTQ | 38.85±1.1 | 16.95±0.17 | 14.04±0.14 | 11.91±0.044 | 10.89±0.045 | 9.868±0.067 |
| | ±Stdev | **BoA** | **34.29±0.39** | **16.09±0.11** | **13.47±0.15** | **11.51±0.094** | **10.55±0.036** | **9.798±0.038** |
| PTB | 0 | GPTQ | 57.62 | 24.77 | 21.53 | 17.56 | 15.68 | 14.56 |
| | | **BoA** | **48.50** | **23.53** | **20.33** | **16.86** | **15.21** | **14.50** |
| | 10 | GPTQ | 55.56 | 25.19 | 21.60 | 17.19 | 15.66 | 14.66 |
| | | **BoA** | **48.37** | **23.35** | **20.27** | **16.71** | **15.13** | **14.50** |
| | 20 | GPTQ | 56.22 | 25.47 | 21.42 | 17.24 | 15.67 | 14.60 |
| | | **BoA** | **47.01** | **23.31** | **20.22** | **16.68** | **15.16** | **14.54** |
| | 50 | GPTQ | 56.68 | 24.94 | 21.44 | 17.36 | 15.74 | 14.52 |
| | | **BoA** | **51.05** | **23.69** | **20.19** | **16.76** | **15.22** | **14.46** |
| | 100 | GPTQ | 52.15 | 25.13 | 21.32 | 17.44 | 15.79 | 14.53 |
| | | **BoA** | **48.66** | **23.50** | **19.93** | **16.77** | **15.21** | **14.46** |
| | Mean | GPTQ | 55.65±2.1 | 25.10±0.26 | 21.46±0.11 | 17.36±0.15 | 15.71±0.057 | 14.57±0.058 |
| | ±Stdev | **BoA** | **48.72±1.5** | **23.48±0.15** | **20.19±0.16** | **16.75±0.068** | **15.19±0.038** | **14.49±0.035** |
| C4 | 0 | GPTQ | 33.90 | 18.18 | 16.10 | 13.60 | 12.62 | 11.76 |
| | | **BoA** | **31.12** | **17.74** | **15.83** | **13.34** | **12.52** | **11.75** |
| | 10 | GPTQ | 34.16 | 18.19 | 16.07 | 13.60 | 12.63 | 11.76 |
| | | **BoA** | **31.72** | **17.72** | **15.70** | **13.34** | **12.52** | **11.74** |
| | 20 | GPTQ | 34.07 | 18.19 | 16.07 | 13.58 | 12.62 | 11.75 |
| | | **BoA** | **31.29** | **17.72** | **15.73** | **13.33** | **12.53** | **11.74** |
| | 50 | GPTQ | 33.68 | 18.16 | 16.06 | 13.60 | 12.67 | 11.76 |
| | | **BoA** | **31.11** | **17.73** | **15.73** | **13.37** | **12.54** | **11.74** |
| | 100 | GPTQ | 33.80 | 18.20 | 16.12 | 13.61 | 12.66 | 11.76 |
| | | **BoA** | **31.38** | **17.75** | **15.70** | **13.36** | **12.55** | **11.75** |
| | Mean | GPTQ | 33.92±0.20 | 18.18±0.015 | 16.08±0.026 | 13.60±0.014 | 12.64±0.023 | 11.76±0.0045 |
| | ±Stdev | **BoA** | **31.32±0.25** | **17.73±0.013** | **15.74±0.053** | **13.35±0.016** | **12.53±0.012** | **11.75±0.0053** |

## D  PSEUDOCODE FOR GPTQ

In this appendix, we provide the pseudocode of the conventional GPTQ (Frantar et al., 2023), which is omitted in the main manuscript due to the page limitation.

---

**Algorithm 2** GPTQ

---

**Input**: weights $\mathbf{W}$, Hessian information $\mathbf{U}_{\text{col}}$, pre-determined step size $\mathbf{S}$, and blocksize $B$

1: **def** GPTQ($\mathbf{W}, \mathbf{U}_{\text{col}}, \mathbf{S}, B = 128$)
2:     Initialize quantized output: $\mathbf{Q} \leftarrow \mathbf{0}_{d_{\text{row}} \times d_{\text{col}}}$
3:     Initialize total quantization errors: $\mathbf{E}_{\text{total}} \leftarrow \mathbf{0}_{d_{\text{row}} \times d_{\text{col}}}$
4:     Initialize block quantization errors: $\mathbf{E}_{\text{block}} \leftarrow \mathbf{0}_{d_{\text{row}} \times B}$
5:     **for** $i = 0, B, 2B, \ldots$ **do**
6:         **for** $j = i, \cdots, i + B - 1$ **do**
7:             Quantize the $j$-th column: $\mathbf{Q}_{:,j} \leftarrow \text{quant}(\mathbf{W}_{:,j}, \mathbf{S})$
8:             Estimate quantization error: $[\mathbf{E}_{\text{block}}]_{:,j-i} \leftarrow (\mathbf{W}_{:,j} - \mathbf{Q}_{:,j})/[\mathbf{U}_{\text{col}}]_{j,j}$
9:             Update weights in block: $\mathbf{W}_{:,j:i+B} \leftarrow \mathbf{W}_{:,j:i+B} - [\mathbf{E}_{\text{block}}]_{:,j-i} \cdot [\mathbf{U}_{\text{col}}]_{j,j:(i+B)}$
10:         **end for**
11:         Update all remaining weights: $\mathbf{W}_{:,i+B:} \leftarrow \mathbf{W}_{:,i+B:} - \mathbf{E}_{\text{block}} \cdot [\mathbf{U}_{\text{col}}]_{i:(i+B),(i+B):}$
12:         Save block quantization errors: $[\mathbf{E}_{\text{total}}]_{:,i:i+B} \leftarrow \mathbf{E}_{\text{block}}$
13:     **end for**

**Output**: quantized weights $\mathbf{Q}$, quantization error $\mathbf{E}_{\text{total}}$

---

