# OpenReview forum: "Attention-aware Post-training Quantization without Backpropagation"
_ICLR.cc/2025/Conference — Submitted to ICLR 2025_

### Official Review · Reviewer_kfPf · 2024-10-29

**Soundness:** 3
**Presentation:** 2
**Contribution:** 2
**Rating:** 3
**Confidence:** 5

**Summary:**

This paper presents a training-free post-training quantization method based on GPTQ. It introduces inter-layer interaction by calculating Hessian matrices using an attention module instead of a simple linear module in LLMs. Additionally, the paper proposes techniques to improve the efficiency of Hessian matrix calculations.

**Strengths:**

1. Introducing inter-layer interaction in a training-free manner is innovative.
2. The paper is well-written.

**Weaknesses:**

1. The experimental setup is somewhat outdated. Additional experiments on newer models, such as LLama-2 and LLama-3, are needed.
2. Although the paper introduces a training-free PTQ method, it may be slower than training-based methods. For example, Table 2 shows that BOA takes 1 hour to quantize 2.7B models, while GPTQ quantizes larger 13B models in only 21 minutes. OmniQuant, a training-based method, requires only ~1.1 hours for 7B models. The paper should provide comprehensive comparisons of quantization times to demonstrate the proposed method's effectiveness.
3. The paper focuses on 2-bit per-channel quantization and mentions that "group-wise parameters result in additional memory costs and processing time during inference." However, weight-only quantization aims to alleviate memory constraints during the decoding stage. Group-wise quantization introduces negligible overhead but significantly improves performance and is a common practice in existing inference engines. Therefore, the paper should include results for group-wise quantization.

**Questions:**

Please refer weaknesses for details.

---

> ### Author Response · Authors · 2024-11-19
>
> We appreciate the reviewer's valuable comments and constructive suggestions on our work.
> Our point-to-point response is as follows.
> Please refer to the end of our final response for the list of references.
>
> **1. Additional experiments on newer models such as LLaMA-2 and LLaMA-3 & Group-wise quantization results**
>
>  - We appreciate the reviewer's suggestion. The main reason why we utilized OPT and BLOOM models in our validation is that the performance comparison on various sizes of models (from 125M to 30B) is possible.
>
>  - As suggested, we have quantized recent LLaMA2 and LLaMA3 models using the proposed BoA and GPTQ (see Table I below). We have also included group-wise quantization results, as suggested by the reviewer. As evident, the proposed BoA uniformly outperforms GPTQ by a significant margin. For almost all quantization configurations, BoA achieves at least 8% improvement over GPTQ in the zero-shot accuracy performance. In particular, the 2-bit quantized "LLaMA2-7B" model obtained by BoA even performs better than the "LLaMA2-13B" model quantized with GPTQ, even applied with group-wise quantization parameters (see the performance of W2G256 LLaMA2-13B obtained by GPTQ).
>
> <Table I. Quantization performance of BoA and GPTQ on LLaMA2 and LLaMA3 models transformed via QuaRot. 'GN' means that quantization has been applied to groups of N consecutive weights.>
>
> (a) Perplexity ($\downarrow$)
>
> |Model|Precision|Method|Wiki2|C4|
> |-|-|-|-|-|
> |LLaMA2-7B|W2|GPTQ|39.56|47.37|
> |||**BoA**|**14.77**|**18.41**|
> ||W2G256|GPTQ|37.63|43.46|
> |||**BoA**|**13.41**|**16.80**|
> ||W2G64|GPTQ|29.38|36.77|
> |||**BoA**|**11.63**|**14.68**|
> ||W2G16|GPTQ|13.75|17.22|
> |||**BoA**|**8.880**|**11.33**|
> |LLaMA2-13B|W2|GPTQ|21.89|27.48|
> |||**BoA**|**11.93**|**18.14**|
> ||W2G256|GPTQ|15.17|19.24|
> |||**BoA**|**10.47**|**13.71**|
> ||W2G64|GPTQ|13.15|17.09|
> |||**BoA**|**9.116**|**11.99**|
> ||W2G16|GPTQ|9.819|13.28|
> |||**BoA**|**7.231**|**9.589**|
> |LLaMA3-8B|W2|GPTQ|40.30|51.92|
> |||**BoA**|**23.50**|**31.47**|
> ||W2G256|GPTQ|34.65|43.50|
> |||**BoA**|**21.41**|**29.09**|
> ||W2G64|GPTQ|25.83|36.04|
> |||**BoA**|**17.85**|**24.81**|
> ||W2G16|GPTQ|15.81|22.98|
> |||**BoA**|**13.00**|**18.90**|
>
> (b) Zero-shot accuracy ($\uparrow$)
>
> |Model|Precision|Method|ARC-c|ARC-e|HellaSwag|Average|
> |-|-|-|-|-|-|-|
> |LLaMA2-7B|W2|GPTQ|22.61|34.81|33.56|30.33|
> |||**BoA**|30.89|55.05|51.22|**45.72**|
> ||W2G256|GPTQ|20.99|35.61|31.89|29.50|
> |||**BoA**|29.61|55.01|52.72|**45.78**|
> ||W2G64|GPTQ|24.40|39.02|36.12|33.18|
> |||**BoA**|33.02|59.85|55.68|**49.52**|
> ||W2G16|GPTQ|27.22|50.21|51.81|43.08|
> |||**BoA**|36.43|63.47|63.13|**54.34**|
> |LLaMA2-13B|W2|GPTQ|25.60|39.31|38.27|34.39|
> |||**BoA**|31.31|58.38|53.07|**47.59**|
> ||W2G256|GPTQ|29.44|51.43|49.58|43.48|
> |||**BoA**|35.67|62.04|59.32|**52.34**|
> ||W2G64|GPTQ|29.78|50.55|50.59|43.64|
> |||**BoA**|37.54|64.39|62.45|**54.79**|
> ||W2G16|GPTQ|35.41|60.10|59.73|51.75|
> |||**BoA**|42.32|69.78|69.04|**60.38**|
> |LLaMA3-8B|W2|GPTQ|24.15|40.24|42.96|35.78|
> |||**BoA**|29.86|51.68|51.85|**44.46**|
> ||W2G256|GPTQ|26.19|45.37|42.57|38.04|
> |||**BoA**|30.46|55.26|52.61|**46.11**|
> ||W2G64|GPTQ|29.01|49.37|45.94|41.44|
> |||**BoA**|32.94|59.64|54.95|**49.18**|
> ||W2G16|GPTQ|34.39|61.91|58.29|51.53|
> |||**BoA**|39.85|65.74|63.17|**56.25**|

---

> > ### Author Response · Authors · 2024-11-19
> >
> > **2. Comprehensive comparisons of quantization times. Although the paper introduces a training-free PTQ method, it may be slower than training-based methods. For example, Table 2 shows that BoA takes 1 hour to quantize 2.7B models, while GPTQ quantizes larger 13B models in only 21 minutes. OmniQuant, a training-based method, requires only 1.1 hours for 7B models.**
> >
> >  - We appreciate the reviewer's comments. In Appendix C.4, we have compared the quantization processing times of the proposed BoA, the conventional training-free method (GPTQ), and training-based methods (OmniQuant and AffineQuant). We note that the processing times of GPTQ and OmniQuant are longer than those reported in the original papers [1], [2] due to the following reasons:
> >
> >    - For GPTQ, we set quantization parameters (scale and zero-point) to minimize the layer-wise reconstruction error (see line 364), which requires a grid search [Section 3.1, 3]. It should be noted that the naive Min-Max-based quantization parameters used in the original GPTQ paper [1] can accelerate the quantization process but result in extremely worse low-bit quantization performance (perplexity is larger than $10^{3}$; see [Table 4, 3]).
> >
> >    - For OmniQuant, the authors reported the processing time of training the model for 20 epochs [Table A12, 2]. However, for the 2-bit quantization, they actually performed training for 40 epochs [Section 4.1, 2]. Therefore, we measure the quantization processing time required to conduct training for 40 epochs.
> >
> >  - As Table 13 in Appendix C.4 shows, GPTQ requires a shorter processing time and a smaller amount of memory than those required by the proposed BoA. This is because GPTQ quantizes all the rows of the weight matrix simultaneously by assuming independence between different layers. In contrast, BoA sequentially quantizes sub-weight matrices (see Figure 1(b)) to consider the inter-layer dependencies within the attention module, which eventually leads to significantly better quantization performance than GPTQ (at least 8% improvement in zero-shot accuracy; see Table I above). Clearly, there is a trade-off between quantization speed / memory cost and accuracy. In real situations, when one needs to preserve the performance of the original model as much as possible, the proposed BoA would be an intriguing solution (see Table I above). It should be noted that such additional processing time is imposed only during the quantization step, and the real inference time of quantized models obtained by BoA is exactly the same as that of GPTQ.
> >
> >  - We emphasize that the proposed BoA performs significantly better than existing training-based approaches (e.g., OmniQuant and AffineQuant), yet facilitates faster quantization (see Tables 6 and 13 in the main text). We note that OmniQuant does not take too much time for quantizing large LLMs even though it performs gradient-based optimization. This is because OmniQuant reduces the number of learnable parameters greatly to accelerate gradient-based optimization. Specifically, instead of optimizing integer weights (which requires a large number of learnable parameters), OmniQuant relies on naive nearest-rounding for assigning integer weights. It learns only a small number of quantization parameters and certain parameters related to the model transformation. In doing so, the quantization process can be accelerated, but OmniQuant suffers from an unstable quantization process or collapses for low-bit quantization (see Table 6 in the main text). It is worth noting that AffineQuant improves OmniQuant by introducing additional learnable parameters [4]. While this enhances quantization performance, it incurs huge processing time compared to the proposed BoA (4 times longer processing time; see Table 13(a)), which demonstrates the inefficiency of training-based approaches over the proposed method.
> >
> > <List of references>
> >
> > [1] E. Frantar et. al., "GPTQ: Accurate post-training quantization for generative pre-trained Transformers," ICLR 2023.
> >
> > [2] W. Shao et. al., "OmniQuant: Omnidirectionally calibrated quantization for large language models," ICLR 2024.
> >
> > [3] Y. Jeon et. al., "A frustratingly easy post-training quantization scheme for LLMs," EMNLP 2023.
> >
> > [4] Y. Ma et. al., "AffineQuant: Affine transformation quantization for large language models," ICLR 2024.

---

> > > ### Author Response · Authors · 2024-11-25
> > >
> > > Dear Reviewer kfPf
> > >
> > > Thanks for your time you dedicated to reviewing our paper!
> > >
> > > You were concerned that we had not provided experimental results on recent models, group-wise quantization results, and comparisons of quantization processing times.
> > >
> > > We think our main rebuttal addresses these concerns due to the following reasons:
> > >  - We have provided quantization results on recent LLaMA2 and LLaMA3 models and group-wise quantization results. Our results demonstrate that for almost all quantization configurations, the proposed BoA achieves **at least 8% improvement over GPTQ** in the zero-shot accuracy performance (see Table I in the main rebuttal). In particular, the 2-bit quantized "LLaMA2-7B" model obtained by BoA even performs better than the "LLaMA2-13B" model quantized with GPTQ, even applied with group-wise quantization parameters (see the performance of W2G256 LLaMA2-13B obtained by GPTQ).
> > >  - We have provided comprehensive comparisons of quantization processing times. Overall, the proposed BoA performs significantly better than existing training-based approaches such as OmniQuant and AffineQuant (see Table 6), yet facilitates faster quantization (e.g., **4 times faster than AffineQuant**; see Table 13(a)).
> > >
> > > If you have any further concerns, please let us know. If not, we would be very grateful if you were to consider increasing your score.

---

> > > > ### Comment · Reviewer_kfPf · 2024-11-26
> > > >
> > > > The comparisons is not enough.
> > > >
> > > > Though the proposed method is training free, the quantization time compared to existing training-based methods (such as AutoRound [1], OmniQuant) is limited. Therefore, it should also include these methods into the comparison table, especially for the practical group-wise quantization.
> > > >
> > > >
> > > > [1] Optimize weight rounding via signed gradient descent for the quantization of llms

---

> > > > > ### Author Response · Authors · 2024-11-27
> > > > >
> > > > > We appreciate the reviewer's comments.
> > > > >
> > > > >  - As suggested, we have compared the group-wise quantization performance of the proposed BoA, OmniQuant [1], and AutoRound [2] (see Table I below).
> > > > >     - For OmniQuant, we have summarized the results reported in the original paper [Table 1, 1].
> > > > >     - For AutoRound, we have run the official code provided by the authors and reported the obtained results.
> > > > >
> > > > >  - **Unstable and unsatisfactory performance of AutoRound**
> > > > >     - AutoRound suffers from an unstable training process due to the gradient approximation involved in the quantization parameter learning (see 'NaN' in [Table 14, 2]), which is similar to OmniQuant (see 'NaN' in Table 6).
> > > > >     - The 2-bit quantization performance of AutoRound collapses (perplexity is larger than $10^{3}$), as reported in the original paper [Section 6, 2].
> > > > >
> > > > >  - **Performance comparison**
> > > > >     - As evident, regardless of whether group-wise quantization is applied, the proposed BoA performs better than OmniQuant and AutoRound.
> > > > >     - In particular, the proposed BoA outperforms OmniQuant and AutoRound by a significant margin for the 2-bit quantization.
> > > > >
> > > > > We believe that these results are sufficient to conclude that the proposed BoA exhibits competitive performance for group-wise quantization as well as standard per-channel quantization. We hope the reviewer finds the above satisfactory. If you have any further concerns, please let us know.
> > > > >
> > > > > <Table I. Group-wise quantization performance (perplexity ($\downarrow$) for WikiText-2) of the proposed BoA, OmniQuant, and AutoRound. 'GN' means that quantization has been applied to groups of N consecutive weights.>
> > > > >
> > > > > |Precision|Method|LLaMA2-7B|LLaMA2-13B|
> > > > > |-|-|-|-|
> > > > > |W2G128|OmniQuant|11.06|8.26|
> > > > > ||AutoRound|19.51|7.91|
> > > > > ||**BoA**|**9.78**|**7.81**|
> > > > > |W2G64|OmniQuant|9.62|7.56|
> > > > > ||AutoRound|17.70|7.51|
> > > > > ||**BoA**|**8.99**|**7.39**|
> > > > > |W2|OmniQuant|37.37|17.21|
> > > > > ||AutoRound|1.0e4|2.6e3|
> > > > > ||**BoA**|**11.04**|**8.94**|
> > > > >
> > > > > <List of references>
> > > > >
> > > > > [1] W. Shao et. al., "OmniQuant: Omnidirectionally calibrated quantization for large language models," ICLR 2024.
> > > > >
> > > > > [2] W. Cheng et. al., "Optimize weight rounding via signed gradient descent for the quantization of LLMs," EMNLP 2024.

---

> > > > > > ### Comment · Reviewer_kfPf · 2024-11-29
> > > > > >
> > > > > > The reported AutoRound results are worse than OmniQuant.
> > > > > >
> > > > > > However, as shown in the official open-source repo of AutoRound [https://github.com/intel/auto-round/blob/main/docs/acc.md](https://github.com/intel/auto-round/blob/main/docs/acc.md), we can see that AutoRound surpass OmniQuant with a large margin, such as +8% Acc. in w2g128 llama-2-7B. I suggest to reuse these results for comparisons.

---

> ### Author Response · Authors · 2024-11-29
>
> We appreciate the reviewer's comments.
>
>  - We kindly ask the reviewer to see the _**official results in Table 14 of the original AutoRound paper**_. For the setting that the reviewer pointed out (W2G128 quantization of LLaMA2-7B), _**AutoRound's perplexity performance has been reported as NaN (which means that the training loss diverges),**_ which contradicts the results that the reviewer mentioned. For this reason, we thought that the reported results were strange, so we ran the official AutoRound code and reported the obtained results.
>
>  - We think that the performance of AutoRound varies significantly with random seeds used to sample calibration data, as can be observed in the table below. As mentioned by the reviewer, _**AutoRound can occasionally outperform OmniQuant (seed 0), but AutoRound shows inferior performance in most cases and even collapses (i.e., perplexity is 96.78) for seed 500**_.
>
> - In contrast, the proposed BoA exhibits much smaller variation with different calibration data, that is, _**the standard deviation (stdev) of BoA is only 0.045 while stdev of AutoRound is 30.55**_. Furthermore, BoA uniformly outperforms AutoRound for all different seeds, which clearly demonstrates the outstanding and stable performance of BoA.
>
> We hope the reviewer finds our response satisfactory. If you have any further concerns, please let us know.
>
> <Table. W2G128 performance of AutoRound and the proposed BoA on LLaMA2-7B (perplexity ($\downarrow$) for WikiText-2)>
>
> |Method|Seed|0|10|20|100|200|500|1000|Average $\pm$ Stdev|
> |-|-|-|-|-|-|-|-|-|-|
> |AutoRound||10.00|15.25|21.58|12.82|24.59|96.78|17.98|**28.43 $\pm$ 30.55**|
> |**BoA**||9.777|9.747|9.747|9.707|9.708|9.647|9.676|**9.716 $\pm$ 0.045**|

---

### Official Review · Reviewer_RmBt · 2024-11-03

**Soundness:** 2
**Presentation:** 3
**Contribution:** 2
**Rating:** 3
**Confidence:** 3

**Summary:**

This paper presents a post-training quantization method called BOA that incorporates inter-layer dependencies without relying on backpropagation. BOA leverages attention-aware Hessian matrices to capture dependencies within the attention module, a relatively rare approach in existing PTQ methods. Additionally, BOA demonstrates compatibility with techniques like SmoothQuant and Z-FOLD, allowing for further enhancements in quantization performance. However, despite these strengths, BOA does not show sufficient memory and processing time benefits compared to existing PTQ methods. The experiments are conducted on outdated models, and the comparison methods lack recent advancements. Adding more experiments with up-to-date models and techniques would strengthen the paper.

**Strengths:**

1.	The paper introduces an innovative PTQ method that cleverly captures inter-layer dependencies within attention modules through attention-aware Hessian matrices while avoiding backpropagation overhead.
2.	BOA is compatible with other techniques, such as SmoothQuant and Z-FOLD, enabling further improvements in quantization accuracy by integrating different quantization strategies.

**Weaknesses:**

1.	The experiments are primarily conducted on BLOOM, LLaMA1, and OPT models, which are somewhat outdated compared to current state-of-the-art models. The paper lacks validation on more recent models, such as the LLaMA3 series.
2.	Although the paper introduces various techniques to reduce computational overhead and claims to use a Hessian-based strategy to avoid time-consuming gradient-based optimization, as shown in Table 13, BOA’s actual overhead in terms of memory and processing time is greater than GPTQ. Additionally, in Tables 3, 4, and 5, even under 2-bit quantization, BOA's improvement over GPTQ is marginal. For Table 6, it’s worth noting that GPTQ can also integrate certain quantization algorithms, like QuaRot [1] and SpinQuant [2], to achieve better results. Including comparisons with these methods is recommended.

[1] Ashkboos S, Mohtashami A, Croci M L, et al. Quarot: Outlier-free 4-bit inference in rotated LLMs. arXiv preprint arXiv:2404.00456, 2024.
[2] Liu Z, Zhao C, Fedorov I, et al. SpinQuant—LLM quantization with learned rotations. arXiv preprint arXiv:2405.16406, 2024.

**Questions:**

1.How does the performance of BOA compare when tested on more advanced models, such as the LLaMA3 series, instead of the relatively outdated models used in the paper?
2.How does BOA's accuracy compare to more recent quantization methods, such as QuaRot and SpinQuant?

---

> ### Author Response · Authors · 2024-11-19
>
> We appreciate the reviewer's valuable comments and constructive suggestions on our work.
> Our point-to-point response is as follows.
>
> **1. Validation on more recent models such as LLaMA3 & Integration with more recent quantization methods such as QuaRot**
>
>  - We appreciate the reviewer's suggestion. The main reason why we utilized OPT and BLOOM models in our validation is that the performance comparison on various sizes of models (from 125M to 30B) is possible.
>
>  - As suggested, we have quantized the recent LLaMA3-8B model with the proposed BoA and the conventional GPTQ. Furthermore, to check the compatibility with the recent transformation method, we have transformed LLaMA3-8B via QuaRot and then measured the quantization performance on the transformed LLaMA3-8B model (see Table I below). We observe that both BoA and GPTQ perform better when QuaRot is applied. As evident, the proposed BoA uniformly performs better than GPTQ. In particular, when QuaRot has been applied, BoA outperforms GPTQ by a significant margin (9% improvement in the zero-shot accuracy).
>
> <Table I. INT2 quantization performance of BoA and GPTQ on LLaMA3-8B>
>
> (a) Perplexity ($\downarrow$)
> |Transformation|Method|Wiki2|C4|
> |-|-|-|-|
> |None|GPTQ|76.77|54.50|
> ||**BoA**|**71.75**|**46.04**|
> |QuaRot|GPTQ|40.30|51.92|
> ||**BoA**|**23.50**|**31.47**|
>
> (b) Zero-shot accuracy ($\uparrow$)
> |Transformation|Method|ARC-c|ARC-e|HellaSwag|Average|
> |-|-|-|-|-|-|
> |None|GPTQ|20.65|32.66|44.00|32.44|
> ||**BoA**|22.70|35.73|47.37|**35.27**|
> |QuaRot|GPTQ|24.15|40.24|42.96|35.78|
> ||**BoA**|29.86|51.68|51.85|**44.46**|
>
> **2. Marginal improvement over GPTQ**
>
>  - We appreciate the reviewer's comment. For a more thorough comparison, we have quantized recent LLaMA2 and LLaMA3 models with the proposed BoA and GPTQ (see Table II below). We also include the group-wise quantization results, as suggested by the reviewer kfPf. As evident, the proposed BoA uniformly outperforms GPTQ for all models. For almost all quantization configurations, BoA achieves at least 8% improvement over GPTQ in the zero-shot accuracy performance. In particular, the 2-bit quantized "LLaMA2-7B" model obtained by BoA even performs better than the "LLaMA2-13B" model quantized with GPTQ, even applied with group-wise quantization parameters (see the performance of W2G256 LLaMA2-13B obtained by GPTQ). In this sense, we believe the improvement over GPTQ is not marginal. We hope the reviewer's kind evaluation and acknowledgment on our effort.
>
> <Table II. Quantization performance of BoA and GPTQ on LLaMA2 and LLaMA3 models transformed via QuaRot. 'GN' means that quantization has been applied to groups of N consecutive weights.>
>
> (a) Perplexity ($\downarrow$)
>
> |Model|Precision|Method|Wiki2|C4|
> |-|-|-|-|-|
> |LLaMA2-7B|W2|GPTQ|39.56|47.37|
> |||**BoA**|**14.77**|**18.41**|
> ||W2G256|GPTQ|37.63|43.46|
> |||**BoA**|**13.41**|**16.80**|
> ||W2G64|GPTQ|29.38|36.77|
> |||**BoA**|**11.63**|**14.68**|
> ||W2G16|GPTQ|13.75|17.22|
> |||**BoA**|**8.880**|**11.33**|
> |LLaMA2-13B|W2|GPTQ|21.89|27.48|
> |||**BoA**|**11.93**|**18.14**|
> ||W2G256|GPTQ|15.17|19.24|
> |||**BoA**|**10.47**|**13.71**|
> ||W2G64|GPTQ|13.15|17.09|
> |||**BoA**|**9.116**|**11.99**|
> ||W2G16|GPTQ|9.819|13.28|
> |||**BoA**|**7.231**|**9.589**|
> |LLaMA3-8B|W2|GPTQ|40.30|51.92|
> |||**BoA**|**23.50**|**31.47**|
> ||W2G256|GPTQ|34.65|43.50|
> |||**BoA**|**21.41**|**29.09**|
> ||W2G64|GPTQ|25.83|36.04|
> |||**BoA**|**17.85**|**24.81**|
> ||W2G16|GPTQ|15.81|22.98|
> |||**BoA**|**13.00**|**18.90**|
>
> (b) Zero-shot accuracy ($\uparrow$)
>
> |Model|Precision|Method|ARC-c|ARC-e|HellaSwag|Average|
> |-|-|-|-|-|-|-|
> |LLaMA2-7B|W2|GPTQ|22.61|34.81|33.56|30.33|
> |||**BoA**|30.89|55.05|51.22|**45.72**|
> ||W2G256|GPTQ|20.99|35.61|31.89|29.50|
> |||**BoA**|29.61|55.01|52.72|**45.78**|
> ||W2G64|GPTQ|24.40|39.02|36.12|33.18|
> |||**BoA**|33.02|59.85|55.68|**49.52**|
> ||W2G16|GPTQ|27.22|50.21|51.81|43.08|
> |||**BoA**|36.43|63.47|63.13|**54.34**|
> |LLaMA2-13B|W2|GPTQ|25.60|39.31|38.27|34.39|
> |||**BoA**|31.31|58.38|53.07|**47.59**|
> ||W2G256|GPTQ|29.44|51.43|49.58|43.48|
> |||**BoA**|35.67|62.04|59.32|**52.34**|
> ||W2G64|GPTQ|29.78|50.55|50.59|43.64|
> |||**BoA**|37.54|64.39|62.45|**54.79**|
> ||W2G16|GPTQ|35.41|60.10|59.73|51.75|
> |||**BoA**|42.32|69.78|69.04|**60.38**|
> |LLaMA3-8B|W2|GPTQ|24.15|40.24|42.96|35.78|
> |||**BoA**|29.86|51.68|51.85|**44.46**|
> ||W2G256|GPTQ|26.19|45.37|42.57|38.04|
> |||**BoA**|30.46|55.26|52.61|**46.11**|
> ||W2G64|GPTQ|29.01|49.37|45.94|41.44|
> |||**BoA**|32.94|59.64|54.95|**49.18**|
> ||W2G16|GPTQ|34.39|61.91|58.29|51.53|
> |||**BoA**|39.85|65.74|63.17|**56.25**|

---

> > ### Author Response · Authors · 2024-11-19
> >
> > **3. BoA's actual overhead in terms of memory and processing time is greater than GPTQ.**
> >
> >  - As mentioned by the reviewer, GPTQ requires a shorter processing time and a smaller amount of memory than those required by the proposed BoA. This is because GPTQ quantizes all the rows of the weight matrix simultaneously by assuming independence between different layers. In contrast, BoA sequentially quantizes sub-weight matrices (see Figure 1(b)) to consider the inter-layer dependencies within the attention module, which eventually leads to significantly better quantization performance than GPTQ (at least 8% improvement in zero-shot accuracy; see Tables I and II above). It should be noted that such additional processing time is imposed only during the quantization step, and the real inference time of quantized models obtained by BoA is exactly the same as that of GPTQ.
> >
> >  - Clearly, there is a trade-off between quantization speed / memory cost and accuracy. In real situations, when one needs to preserve the performance of the original model as much as possible, the proposed BoA would be an intriguing solution (see Tables I and II above). Furthermore, we emphasize that the proposed BoA performs significantly better than existing gradient-based approaches (e.g., OmniQuant and AffineQuant), yet facilitates faster quantization (see Tables 6 and 13 in the main text).
> >
> >  - Even when the memory resource is limited, the proposed BoA can be used with some relaxation. Specifically, we note that the large memory cost of BoA for hyper-scale LLMs (e.g., 13B and 30B) is attributable to the row-wise Hessian for the value projection ($\mathbf{X} \mathbf{A} _{h} ^{T} \mathbf{A} _{h} \mathbf{X} ^{T}$; see Eq. (12)) whose shape is $H \times d \times d$ ($H$ is the number of attention heads and $d$ is the embedding dimension). In memory-limited cases, we can mitigate the memory cost of BoA by considering inter-layer dependencies only for query and key projections and applying the standard Hessian ($\mathbf{X} \mathbf{X} ^{T}$) for the value projection. Indeed, when considering only query and key projections, BoA requires almost same amount of memory as GPTQ, yet still exhibiting better performance (see Table 14 in the main text).

---

> > > ### Comment · Reviewer_RmBt · 2024-11-22
> > > **I maintain my concerns about this paper's positioning and contributions.**
> > >
> > > I maintain my concerns about this paper's positioning and contributions. The authors appear to deliberately distance themselves from established PTQ methods like SpinQuant, AffineQuant, and OmniQuant, choosing to compare primarily with GPTQ. This approach is problematic for several reasons:
> > >
> > > 1.Regarding efficiency, PTQ methods have gained widespread adoption precisely because of their minimal computational and data requirements. The authors themselves acknowledge that methods like OmniQuant require few tuning parameters. The paper fails to demonstrate significant advantages in this crucial aspect of quantization.
> > >
> > > 2.Regarding accuracy, the transformations employed in the referenced PTQ methods are invertible (e.g., affine transformations in SpinQuant and AffineQuant, scale-shift transformations in OmniQuant). These transformations can theoretically be absorbed into subsequent layers - for instance, if Layer 1 applies a transformation to W, Layer 2 can apply an inverse transformation to maintain distributional consistency. This mathematical property contributes to PTQ's strong generalization capabilities, which has led to its widespread adoption in both industry and academia.
> > >
> > > Given these considerations, I fail to see substantial advantages or meaningful differentiation in the proposed method. The paper does not present convincing evidence to support its claims of superiority over existing approaches. Therefore, I maintain my original assessment and score.

---

> > > > ### Author Response · Authors · 2024-11-22
> > > >
> > > > We sincerely appreciate the reviewer's further comments. Our point-to-point responses are as follows.
> > > >
> > > > **1. Positioning and contributions of our work**
> > > >  - We kindly note that the proposed method is orthogonal to the transformation-based approaches that the reviewer mentioned. Specifically, recent PTQ methods for the LLM quantization can be classified into two orthogonal categories.
> > > >     - methods that optimize integer weights based on approximated Hessian matrices (e.g., GPTQ)
> > > >     - methods that transform a model into a more quantization-favorable form (e.g., SmoothQuant [1], QuIP [2], Z-Fold [3], OmniQuant [4], AffineQuant [5], QuaRot [6], and SpinQuant [7])
> > > >  - We emphasize that the proposed method is the integer weight optimization method, so we chose to compare primarily with GPTQ.
> > > >  - It is worth noting that, similar to GPTQ, which can be integrated with transformation-based methods [2], [3], [6], [7], the proposed approach can also be combined with existing techniques to improve their performance. Indeed, our results in Table 5 (in the main text) and Table I (see below) demonstrate that the quantization performance can be boosted by combining the proposed method with existing transformation methods such as SmoothQuant, Z-Fold, and QuaRot. We note that the reason why we chose SmoothQuant, Z-Fold, and QuaRot in our integration is because those methods do not rely on backpropagation, unlike other methods (OmniQuant, AffineQuant, and SpinQuant) that depend on gradient-based optimization.
> > > >
> > > > <Table I. INT2 quantization performance of BoA and GPTQ on LLaMA3-8B>
> > > >
> > > > (a) Perplexity ($\downarrow$)
> > > > |Transformation|Method|Wiki2|C4|
> > > > |-|-|-|-|
> > > > |None|GPTQ|76.77|54.50|
> > > > ||**BoA**|**71.75**|**46.04**|
> > > > |QuaRot|GPTQ|40.30|51.92|
> > > > ||**BoA**|**23.50**|**31.47**|
> > > >
> > > > (b) Zero-shot accuracy ($\uparrow$)
> > > > |Transformation|Method|ARC-c|ARC-e|HellaSwag|Average|
> > > > |-|-|-|-|-|-|
> > > > |None|GPTQ|20.65|32.66|44.00|32.44|
> > > > ||**BoA**|22.70|35.73|47.37|**35.27**|
> > > > |QuaRot|GPTQ|24.15|40.24|42.96|35.78|
> > > > ||**BoA**|29.86|51.68|51.85|**44.46**|
> > > >
> > > > **2. Accuracy improvement over existing transformation-based methods**
> > > >  - We understand the benefits of existing transformation-based methods that the reviewer mentioned.
> > > >  - However, transformation-based methods, such as OmniQuant and AffineQuant, perform significantly worse than the proposed method, especially for low-bit quantization (see Table 6), because they rely on the naive nearest-rounding when assigning integer weights. **For example, the perplexity of OmniQuant and AffineQuant is larger than $10^{3}$ in some cases while the proposed BoA exhibits reasonable perplexity across all sizes of models (see Table 6).**
> > > >  - Moreover, **OmniQuant and AffineQuant suffer from an unstable quantization process** due to the gradient approximation involved in the quantization parameter learning **(see 'NaN' in Table 6)**.
> > > >  - Furthermore, we have newly compared the quantization performance of the proposed method, OmniQuant, and AffineQuant on LLaMA2 models (see Table II below). We note that LLaMA3 is excluded from our comparison because the official codes of OmniQuant and AffineQuant do not support the quantization of LLaMA3 models. Our results in Table 6 and Table II validate that the proposed method outperforms OmniQuant and AffineQuant, with respect to both perplexity and zero-shot accuracy performance.
> > > >
> > > > <Table II. INT2 quantization performance of BoA, GPTQ, and existing transformation-based methods on LLaMA2 models. 'NaN' means that loss diverges in the quantization process>
> > > >
> > > > |Model|Method|Wiki2($\downarrow$)|ARC-c|ARC-e|HellaSwag|Average($\uparrow$)|
> > > > |-|-|-|-|-|-|-|
> > > > |LLaMA2-7B|GPTQ|39.56|22.61|34.81|33.56|30.33|
> > > > ||OmniQuant|35.40|25.00|38.80|42.97|35.59|
> > > > ||AffineQuant|NaN|NaN|NaN|NaN|NaN|
> > > > ||**BoA**|**14.77**|30.89|55.05|51.22|**45.72**|
> > > > |LLaMA2-13B|GPTQ|21.89|25.60|39.31|38.27|34.39|
> > > > ||OmniQuant|20.19|27.13|47.98|53.27|42.79|
> > > > ||AffineQuant|18.49|30.80|52.90|57.74|47.15|
> > > > ||**BoA**|**11.93**|31.31|58.38|53.07|**47.59**|

---

> > > > ### Author Response · Authors · 2024-11-22
> > > >
> > > > **3. Efficiency over OmniQuant and AffineQuant**
> > > >  - In Appendix C.4, we compared the quantization processing times of the proposed BoA, OmniQuant, and AffineQuant. We note that we used the same number of GPUs and the same amount of calibration data across all quantization methods.
> > > >  - From Table 13(a), we observe that although OmniQuant and AffineQuant do not optimize integer weights, their processing time is still longer than that required by the proposed method because they rely on time-intensive gradient-based optimization. In particular, AffineQuant requires 4 times longer processing time than that required by the proposed method; **for example, AffineQuant needs 18.41 hours and 44.25 hours for quantizing 13B and 30B models, respectively, while the proposed method can finish quantization in 5 hours and 11 hours, respectively.**
> > > >
> > > > Due to such improvement in accuracy and efficiency together with the compatibility with existing transformation-based methods, we believe that the contribution of this work is meaningful and valuable. We hope for the reviewer's kind evaluation and recognition of our effort.
> > > >
> > > > <List of references>
> > > >
> > > > [1] G. Xiao et. al., "SmoothQuant: Accurate and efficient post-training quantization for large language models," ICML 2023.
> > > >
> > > > [2] J. Chee et. al., "QuIP: 2-bit quantization of large language models with guarantees," NeurIPS 2023.
> > > >
> > > > [3] Y. Jeon et. al., "A frustratingly easy post-training quantization scheme for LLMs," EMNLP 2023.
> > > >
> > > > [4] W. Shao et. al., "OmniQuant: Omnidirectionally calibrated quantization for large language models," ICLR 2024.
> > > >
> > > > [5] Y. Ma et. al., "AffineQuant: Affine transformation quantization for large language models," ICLR 2024.
> > > >
> > > > [6] S. Ashkboos et. al., "QuaRot: Outlier-free 4-bit inference in rotated LLMs," NeurIPS 2024.
> > > >
> > > > [7] Z. Liu et. al., "SpinQuant: LLM quantization with learned rotations," arXiv 2024.

---

> > > > > ### Author Response · Authors · 2024-11-25
> > > > >
> > > > > Dear Reviewer RmBt
> > > > >
> > > > > Thanks for your time you dedicated to reviewing our paper!
> > > > >
> > > > > You were concerned about our paper's positioning and contributions, and accuracy and efficiency improvement over existing transformation-based PTQ methods.
> > > > >
> > > > > We think our main rebuttal addresses these concerns due to the following reasons:
> > > > >  - We have emphasized that the proposed method aims to **optimize integer weights by capturing inter-layer dependencies without relying on time-intensive, gradient-based optimization**. We have also noted that the proposed method is **orthogonal to existing transformation-based PTQ methods** and **can be integrated with them**. Our results in Tables 5 and 6 and Table I (in the main rebuttal) demonstrate that the proposed method can be used to boost the performance of existing transformation-based methods such as SmoothQuant, Z-Fold, and QuaRot.
> > > > >  - Our results in Table 6 and Table II (in the main rebuttal) clearly demonstrate that the proposed BoA significantly outperforms transformation-based methods such as OmniQuant and AffineQuant. For example, **the perplexity of OmniQuant and AffineQuant is larger than $10^{3}$ in some cases while the proposed BoA exhibits reasonable perplexity across all sizes of models** (see Table 6). Furthermore, OmniQuant and AffineQuant suffer from an unstable quantization process, i.e., **the training loss diverges** (see NaN in Table 6).
> > > > >  - Our results in Table 13(a) clearly demonstrate that the proposed method completes quantization faster than OmniQuant and AffineQuant. In particular, **AffineQuant needs 18.41 hours and 44.25 hours for quantizing 13B and 30B models, respectively, while the proposed method can finish quantization in 5 hours and 11 hours, respectively.**
> > > > >
> > > > > If you have any further concerns, please let us know. If not, we would be very grateful if you were to consider increasing your score.

---

### Official Review · Reviewer_XkiM · 2024-11-03

**Soundness:** 2
**Presentation:** 1
**Contribution:** 2
**Rating:** 3
**Confidence:** 5

**Summary:**

The paper introduced the BOA post-training quantization algorithm designed for LLMs that overcomes the limitations of traditional quantization methods, which struggle with inter-layer dependencies and backpropagation requirements in LLMs. BOA leveraged attention-aware Hessian matrices to better capture inter-layer interactions within the attention module, enhancing performance, especially at low bit-widths. The algorithm employed Hessian relaxation and head-wise simultaneous quantization, to attempt to reduce computational and memory costs, making it feasible for quantizing LLMs without backpropagation.

**Strengths:**

The topic of this paper is of significant importance and represents one of the most active and rapidly evolving research areas in the field. As LLMs grow increasingly complex, their deployment on resource-constrained devices requires innovative solutions to reduce computational and memory demands. Quantization, as a compression technique, has gained considerable traction for enabling efficient deployment of LLMs without sacrificing model accuracy.

**Weaknesses:**

The technical approach of this paper is relatively straightforward, lacking intricate or highly novel methodologies. Additionally, certain English terminology within the paper is used imprecisely, which may affect clarity and readability. The comparison methods are somewhat limited, providing a narrow benchmark for evaluating the proposed technique. Moreover, while the experimental results demonstrate some improvements, the advantage over existing methods is not substantial, suggesting the need for further validation, such as, SmoothQuant, LLMC,QuIP etc.

**Questions:**

The advantage over existing methods is not substantial, suggesting the need for further validation, such as, SmoothQuant, LLMC,QuIP etc.

---

> ### Author Response · Authors · 2024-11-19
>
> We appreciate the reviewer's valuable comments and constructive suggestions on our work.
> Our point-to-point response is as follows.
> Please refer to the end of our final response for the list of references.
>
> **1. Limited comparison, suggesting the need for further validation such as SmoothQuant, LLMC, and QuIP**
>
>  - We appreciate the reviewer's invaluable comments. First, we mention that the proposed method is orthogonal to the approaches that the reviewer mentioned. Specifically, recent LLM quantization methods can be classified into two orthogonal categories:
>
>    - methods that optimize integer weights based on approximated Hessian matrices (e.g., GPTQ)
>
>    - methods that transform a model into a more quantization-favorable form (e.g., SmoothQuant [1], QuIP [2], Z-Fold [3], OmniQuant [4], AffineQuant [5], and QuaRot [6])
>
>  - We note that the proposed BoA is the integer weight optimization method, so we chose GPTQ as our baseline algorithm in our comparison. It should be noted that the proposed BoA can be used to enhance the performance of existing transformation methods. Indeed, our results in Table 5 demonstrate that the quantization performance can be boosted by combining BoA with existing transformation methods such as SmoothQuant and Z-Fold.
>
>  - As suggested by the reviewer, we have newly measured the performance of BoA combined with the recent state-of-the-art transformation method QuaRot [6], which is included in the LLMC toolkit [7] (see Table I below). We observe that both BoA and GPTQ perform better when QuaRot is applied. As evident, the proposed BoA uniformly performs better than GPTQ. In particular, when QuaRot has been applied, BoA outperforms GPTQ by a significant margin (9% improvement in the zero-shot accuracy).
>
>  - We note that we have excluded the comparison with QuIP because QuIP requires additional inference time and memory costs in the real inference stage. Specifically, in QuIP, the weight matrix $\mathbf{W}$ is multiplied by random orthogonal matrices to suppress outliers within weights (i.e., $\mathbf{W} \leftarrow \mathbf{U} \mathbf{W} \mathbf{V} ^{T}$ where $\mathbf{U}$ and $\mathbf{V}$ are random orthogonal matrices; see line 5 in [2, Algorithm 1]). While this technique (called incoherent processing) can suppress outliers, additional post-processing is needed to recover quantized weights ($\widehat{\mathbf{W}} \leftarrow \mathbf{U} ^{T} \widehat{\mathbf{W}} \mathbf{V}$; see line 3 in [2, Algorithm 2]). Such post-processing should be done in the real inference stage, thereby incurring additional inference time and memory costs for storing orthogonal matrices $\mathbf{U}$ and $\mathbf{V}$. To accelerate the post-processing, one can utilize some special hardware or develop dedicated kernels, but unlike server-grade GPUs (e.g. NVIDIA A100), on-device NPUs (e.g. Qualcomm Hexagon) lack support for such additional processing, and customizing kernels for desired functionalities is very challenging.
>
> <Table I. INT2 quantization performance of BoA and GPTQ on LLaMA3-8B>
>
> (a) Perplexity ($\downarrow$)
> |Transformation|Method|Wiki2|C4|
> |-|-|-|-|
> |None|GPTQ|76.77|54.50|
> ||**BoA**|**71.75**|**46.04**|
> |QuaRot|GPTQ|40.30|51.92|
> ||**BoA**|**23.50**|**31.47**|
>
> (b) Zero-shot accuracy ($\uparrow$)
> |Transformation|Method|ARC-c|ARC-e|HellaSwag|Average|
> |-|-|-|-|-|-|
> |None|GPTQ|20.65|32.66|44.00|32.44|
> ||**BoA**|22.70|35.73|47.37|**35.27**|
> |QuaRot|GPTQ|24.15|40.24|42.96|35.78|
> ||**BoA**|29.86|51.68|51.85|**44.46**|

---

> > ### Author Response · Authors · 2024-11-19
> >
> > **2. Advantage over existing methods is not substantial.**
> >
> >  -  We appreciate the reviewer's comment. For a more thorough comparison, we have quantized recent LLaMA2 and LLaMA3 models with the proposed BoA and GPTQ (see Table II below). We also include the group-wise quantization results, as suggested by the reviewer kfPf. As evident, the proposed BoA uniformly outperforms GPTQ for all models. For almost all quantization configurations, BoA achieves at least 8% improvement over GPTQ in the zero-shot accuracy performance. In particular, the 2-bit quantized "LLaMA2-7B" model obtained by BoA even performs better than the "LLaMA2-13B" model quantized with GPTQ, even applied with group-wise quantization parameters (see the performance of W2G256 LLaMA2-13B obtained by GPTQ). In this sense, we believe the improvement over GPTQ is not marginal.
> >
> > <Table II. Quantization performance of BoA and GPTQ on LLaMA2 and LLaMA3 models transformed via QuaRot. 'GN' means that quantization has been applied to groups of N consecutive weights.>
> >
> > (a) Perplexity ($\downarrow$)
> >
> > |Model|Precision|Method|Wiki2|C4|
> > |-|-|-|-|-|
> > |LLaMA2-7B|W2|GPTQ|39.56|47.37|
> > |||**BoA**|**14.77**|**18.41**|
> > ||W2G256|GPTQ|37.63|43.46|
> > |||**BoA**|**13.41**|**16.80**|
> > ||W2G64|GPTQ|29.38|36.77|
> > |||**BoA**|**11.63**|**14.68**|
> > ||W2G16|GPTQ|13.75|17.22|
> > |||**BoA**|**8.880**|**11.33**|
> > |LLaMA2-13B|W2|GPTQ|21.89|27.48|
> > |||**BoA**|**11.93**|**18.14**|
> > ||W2G256|GPTQ|15.17|19.24|
> > |||**BoA**|**10.47**|**13.71**|
> > ||W2G64|GPTQ|13.15|17.09|
> > |||**BoA**|**9.116**|**11.99**|
> > ||W2G16|GPTQ|9.819|13.28|
> > |||**BoA**|**7.231**|**9.589**|
> > |LLaMA3-8B|W2|GPTQ|40.30|51.92|
> > |||**BoA**|**23.50**|**31.47**|
> > ||W2G256|GPTQ|34.65|43.50|
> > |||**BoA**|**21.41**|**29.09**|
> > ||W2G64|GPTQ|25.83|36.04|
> > |||**BoA**|**17.85**|**24.81**|
> > ||W2G16|GPTQ|15.81|22.98|
> > |||**BoA**|**13.00**|**18.90**|
> >
> > (b) Zero-shot accuracy ($\uparrow$)
> >
> > |Model|Precision|Method|ARC-c|ARC-e|HellaSwag|Average|
> > |-|-|-|-|-|-|-|
> > |LLaMA2-7B|W2|GPTQ|22.61|34.81|33.56|30.33|
> > |||**BoA**|30.89|55.05|51.22|**45.72**|
> > ||W2G256|GPTQ|20.99|35.61|31.89|29.50|
> > |||**BoA**|29.61|55.01|52.72|**45.78**|
> > ||W2G64|GPTQ|24.40|39.02|36.12|33.18|
> > |||**BoA**|33.02|59.85|55.68|**49.52**|
> > ||W2G16|GPTQ|27.22|50.21|51.81|43.08|
> > |||**BoA**|36.43|63.47|63.13|**54.34**|
> > |LLaMA2-13B|W2|GPTQ|25.60|39.31|38.27|34.39|
> > |||**BoA**|31.31|58.38|53.07|**47.59**|
> > ||W2G256|GPTQ|29.44|51.43|49.58|43.48|
> > |||**BoA**|35.67|62.04|59.32|**52.34**|
> > ||W2G64|GPTQ|29.78|50.55|50.59|43.64|
> > |||**BoA**|37.54|64.39|62.45|**54.79**|
> > ||W2G16|GPTQ|35.41|60.10|59.73|51.75|
> > |||**BoA**|42.32|69.78|69.04|**60.38**|
> > |LLaMA3-8B|W2|GPTQ|24.15|40.24|42.96|35.78|
> > |||**BoA**|29.86|51.68|51.85|**44.46**|
> > ||W2G256|GPTQ|26.19|45.37|42.57|38.04|
> > |||**BoA**|30.46|55.26|52.61|**46.11**|
> > ||W2G64|GPTQ|29.01|49.37|45.94|41.44|
> > |||**BoA**|32.94|59.64|54.95|**49.18**|
> > ||W2G16|GPTQ|34.39|61.91|58.29|51.53|
> > |||**BoA**|39.85|65.74|63.17|**56.25**|

---

> > > ### Author Response · Authors · 2024-11-19
> > >
> > > **3. Straightforward approach, lacking intricate or highly novel methodologies.**
> > >
> > >  - We appreciate the reviewer's comment. As the reviewers RmBt and kfPf acknowledged, we believe that our contribution is innovative in the sense that the proposed BoA is the first quantization method that attempts to capture inter-layer dependencies **without backpropagation**.
> > >
> > >  - While it is well-known that capturing inter-layer dependencies is beneficial for quantization, all the existing works rely on time-consuming gradient-based optimization [4], [5], [8], which would not be suitable for real-world deployment where models to be deployed are frequently updated and multiple times of hyper-parameter searches are needed. Indeed, the first PTQ method that attempts to capture inter-layer dependencies (called BRECQ [8]) needs more than 10 hours even for relatively small-sized models (e.g., OPT-1.3B), and requires multiple GPU resources to quantize LLMs having more than 7B parameters.
> > >
> > >  - Recently, OmniQuant [4] and AffineQuant [5] accelerated the quantization processing time by learning only a small number of quantization parameters (scale and zero-point) and certain parameters related to the model transformation. However, they suffer from an unstable quantization process due to the gradient approximation involved in the quantization parameter learning and sacrifice the low-bit performance because they apply the naive nearest-rounding when assigning integer weights (see Table 6). Furthermore, although OmniQuant and AffineQuant do not optimize integer weights, their processing time is still longer (e.g., 4 times longer for AffineQuant) than that required by the proposed BoA (see Table 13(a)).
> > >
> > >  - To avoid the aforementioned disadvantages, we established the attention-aware Hessians, which is the first work to consider inter-layer dependencies while circumventing gradient-based optimization. Moreover, we presented several relaxation techniques, without which multiple GPU resources are required and the quantization cannot be done in a reasonable processing time. Due to these reasons, we believe that the contribution of this work is meaningful and valuable. We hope the reviewer's kind evaluation and acknowledgment on our effort to develop a practical quantization solution that captures inter-layer dependencies.
> > >
> > > <List of references>
> > >
> > > [1] G. Xiao et. al., "SmoothQuant: Accurate and efficient post-training quantization for large language models," ICML 2023.
> > >
> > > [2] J. Chee et. al., "QuIP: 2-bit quantization of large language models with guarantees," NeurIPS 2023.
> > >
> > > [3] Y. Jeon et. al., "A frustratingly easy post-training quantization scheme for LLMs," EMNLP 2023.
> > >
> > > [4] W. Shao et. al., "OmniQuant: Omnidirectionally calibrated quantization for large language models," ICLR 2024.
> > >
> > > [5] Y. Ma et. al., "AffineQuant: Affine transformation quantization for large language models," ICLR 2024.
> > >
> > > [6] S. Ashkboos et. al., "QuaRot: Outlier-free 4-bit inference in rotated LLMs," NeurIPS 2024.
> > >
> > > [7] R. Gong et. al., "LLMC: Benchmarking large language model quantization with a versatile compression toolkit," EMNLP 2024.
> > >
> > > [8] Y. Li et. al., "BRECQ: Pushing the limit of post-training quantization by block reconstruction," ICLR 2021.

---

> > > > ### Author Response · Authors · 2024-11-25
> > > >
> > > > Dear Reviewer XkiM
> > > >
> > > > Thanks for your time you dedicated to reviewing our paper!
> > > >
> > > > You were concerned about our paper's marginal improvement, further validation on recent methods, and limited contributions.
> > > >
> > > > We think our main rebuttal addresses these concerns due to the following reasons:
> > > >  - We have provided integration results with recent transformation-based method (named QuaRot), which is included in the LLMC toolkit that the reviewer suggested, and group-wise quantization results. Our results demonstrate that for almost all quantization configurations, **the proposed BoA achieves at least 8% improvement** over GPTQ in the zero-shot accuracy performance (see Tables I and II in the main rebuttal), which we believe is a significant advancement. In particular, the 2-bit quantized "LLaMA2-7B" model obtained by BoA even performs better than the "LLaMA2-13B" model quantized with GPTQ, even applied with group-wise quantization parameters (see the performance of W2G256 LLaMA2-13B obtained by GPTQ).
> > > >  - We have emphasized that the proposed method is the **first to capture inter-layer dependencies without backpropagation**. Existing methods that attempt to capture inter-layer dependencies rely on time-intensive, gradient-based optimization, which results in much longer quantization processing time. For example, **AffineQuant needs 18.41 hours and 44.25 hours for quantizing 13B and 30B models, respectively, while the proposed method can finish quantization in 5 hours and 11 hours, respectively.**
> > > >
> > > > If you have any further concerns, please let us know. If not, we would be very grateful if you were to consider increasing your score.

---

### Official Review · Reviewer_gH6w · 2024-11-04

**Soundness:** 3
**Presentation:** 3
**Contribution:** 3
**Rating:** 6
**Confidence:** 4

**Summary:**

This paper presents a novel post-training quantization (PTQ) method, termed BOA (Backpropagation-free Optimization for Attention-aware PTQ), targeting large language models (LLMs) without relying on backpropagation. The approach introduces attention-aware Hessian matrices that capture inter-layer dependencies within the attention module, aiming to improve quantization accuracy, especially at low bit-widths (e.g., INT2). BOA incorporates techniques like Hessian relaxation and efficient computation of inverse Hessians to mitigate the high computational costs. The method is benchmarked against existing PTQ approaches on LLMs, demonstrating improved performance in terms of perplexity and zero-shot task accuracy.

**Strengths:**

1. The proposed BOA consider inter-layer dependencies within the attention module when optimize a weight-rounding mechanism. It is beneficial to maintain higher quantization accuracy, especially at low-bit precision.

2. The proposed BOA method demonstrates impressive results, especially in the low-bit regime (e.g., INT2 quantization).

3. The paper includes extensive experiments across multiple model types and sizes, demonstrating  scalability across LLMs of different parameter counts.

**Weaknesses:**

1. Novelty Limitations: The primary contribution, the attention-aware Hessian matrix, is an incremental improvement over existing Hessian-based PTQ methods. While capturing inter-layer dependencies within the attention module is beneficial, the idea is not a novel quantization paradigm.

2. The authors introduce optimizations approaches like Hessian relaxation and efficient computation of inverse Hessians, but the results did not show the effect of these optimization methods.

**Questions:**

Refer to 2 in weakness. What is the effectiveness of proposed approaches in terms of efficiency?

---

> ### Author Response · Authors · 2024-11-19
>
> We appreciate the reviewer's valuable comments and constructive suggestions on our work.
> Our point-to-point response is as follows.
> Please refer to the end of our final response for the list of references.
>
> **1. Limited novelty**
>
>  - We appreciate the reviewer's comment. As the reviewers RmBt and kfPf acknowledged, we believe that our contribution is innovative in the sense that the proposed BoA is the first quantization method that attempts to capture inter-layer dependencies **without backpropagation**.
>
>  - While it is well-known that capturing inter-layer dependencies is beneficial for quantization, all the existing works rely on time-consuming gradient-based optimization [1], [2], [3], which would not be suitable for real-world deployment where models to be deployed are frequently updated and multiple times of hyper-parameter searches are needed. Indeed, the first PTQ method that attempts to capture inter-layer dependencies (called BRECQ [1]) needs more than 10 hours even for relatively small-sized models (e.g., OPT-1.3B), and requires multiple GPU resources to quantize LLMs having more than 7B parameters.
>
>  - Recently, OmniQuant [2] and AffineQuant [3] accelerated the quantization processing time by learning only a small number of quantization parameters (scale and zero-point) and certain parameters related to the model transformation. However, they suffer from an unstable quantization process due to the gradient approximation involved in the quantization parameter learning and sacrifice the low-bit performance because they apply the naive nearest-rounding when assigning integer weights (see Table 6). Furthermore, although OmniQuant and AffineQuant do not optimize integer weights, their processing time is still longer (e.g., 4 times longer for AffineQuant) than that required by the proposed BoA (see Table 13(a)).
>
>  - To avoid the aforementioned disadvantages, we established the attention-aware Hessians, which is the first work to consider inter-layer dependencies while circumventing gradient-based optimization. We emphasize that existing Hessian-based PTQ methods, such as GPTQ, cannot capture inter-layer dependencies, which results in significantly worse performance than the proposed method (see Table I for comparison on recent LLMs such as LLaMA2 and LLaMA3). Moreover, we presented several relaxation techniques, without which multiple GPU resources are required and the quantization cannot be done in a reasonable processing time; for more details, please refer to our response to the next comment. Due to these reasons, we believe that the contribution of this work is meaningful and valuable. We hope the reviewer's kind evaluation and acknowledgment on our effort to develop a practical quantization solution that captures inter-layer dependencies.
>
> <Table I. Quantization performance of BoA and GPTQ on LLaMA2 and LLaMA3 models transformed via QuaRot. 'GN' means that quantization has been applied to groups of N consecutive weights.>
>
> (a) Perplexity ($\downarrow$)
>
> |Model|Precision|Method|Wiki2|C4|
> |-|-|-|-|-|
> |LLaMA2-7B|W2|GPTQ|39.56|47.37|
> |||**BoA**|**14.77**|**18.41**|
> ||W2G256|GPTQ|37.63|43.46|
> |||**BoA**|**13.41**|**16.80**|
> ||W2G64|GPTQ|29.38|36.77|
> |||**BoA**|**11.63**|**14.68**|
> ||W2G16|GPTQ|13.75|17.22|
> |||**BoA**|**8.880**|**11.33**|
> |LLaMA2-13B|W2|GPTQ|21.89|27.48|
> |||**BoA**|**11.93**|**18.14**|
> ||W2G256|GPTQ|15.17|19.24|
> |||**BoA**|**10.47**|**13.71**|
> ||W2G64|GPTQ|13.15|17.09|
> |||**BoA**|**9.116**|**11.99**|
> ||W2G16|GPTQ|9.819|13.28|
> |||**BoA**|**7.231**|**9.589**|
> |LLaMA3-8B|W2|GPTQ|40.30|51.92|
> |||**BoA**|**23.50**|**31.47**|
> ||W2G256|GPTQ|34.65|43.50|
> |||**BoA**|**21.41**|**29.09**|
> ||W2G64|GPTQ|25.83|36.04|
> |||**BoA**|**17.85**|**24.81**|
> ||W2G16|GPTQ|15.81|22.98|
> |||**BoA**|**13.00**|**18.90**|
>
> (b) Zero-shot accuracy ($\uparrow$)
>
> |Model|Precision|Method|ARC-c|ARC-e|HellaSwag|Average|
> |-|-|-|-|-|-|-|
> |LLaMA2-7B|W2|GPTQ|22.61|34.81|33.56|30.33|
> |||**BoA**|30.89|55.05|51.22|**45.72**|
> ||W2G256|GPTQ|20.99|35.61|31.89|29.50|
> |||**BoA**|29.61|55.01|52.72|**45.78**|
> ||W2G64|GPTQ|24.40|39.02|36.12|33.18|
> |||**BoA**|33.02|59.85|55.68|**49.52**|
> ||W2G16|GPTQ|27.22|50.21|51.81|43.08|
> |||**BoA**|36.43|63.47|63.13|**54.34**|
> |LLaMA2-13B|W2|GPTQ|25.60|39.31|38.27|34.39|
> |||**BoA**|31.31|58.38|53.07|**47.59**|
> ||W2G256|GPTQ|29.44|51.43|49.58|43.48|
> |||**BoA**|35.67|62.04|59.32|**52.34**|
> ||W2G64|GPTQ|29.78|50.55|50.59|43.64|
> |||**BoA**|37.54|64.39|62.45|**54.79**|
> ||W2G16|GPTQ|35.41|60.10|59.73|51.75|
> |||**BoA**|42.32|69.78|69.04|**60.38**|
> |LLaMA3-8B|W2|GPTQ|24.15|40.24|42.96|35.78|
> |||**BoA**|29.86|51.68|51.85|**44.46**|
> ||W2G256|GPTQ|26.19|45.37|42.57|38.04|
> |||**BoA**|30.46|55.26|52.61|**46.11**|
> ||W2G64|GPTQ|29.01|49.37|45.94|41.44|
> |||**BoA**|32.94|59.64|54.95|**49.18**|
> ||W2G16|GPTQ|34.39|61.91|58.29|51.53|
> |||**BoA**|39.85|65.74|63.17|**56.25**|

---

> > ### Author Response · Authors · 2024-11-19
> >
> > **2. The authors introduce optimization approaches like Hessian relaxation and efficient computation of inverse Hessians, but the results did not show the effect of these optimization methods.**
> >
> >  - We appreciate the reviewer's constructive suggestion. To say conclusion first, without each component developed to simplify the quantization process, the proposed method cannot finish quantization in a reasonable time with a single GPU.
> >
> >  - Without the proposed relaxation on Hessians, we need to compute and store the Jacobian matrix $\mathbf{J}_{\sigma}$ for the softmax function (see Eqs. (8) and (11)). Because the shape of $\mathbf{J} _{\sigma}$ is $H \times L \times L \times L$ where $H$ is the number of attention heads and $L$ is the input sequence length, storing $\mathbf{J} _{\sigma}$ requires more than 400 GB memory even for OPT-125M ($H=12$ and $L=2048$), which is not possible with a single A100 GPU of 80 GB memory.
> >
> >  - Without the proposed efficient computation of inverse Hessians, we need to compute the inverse matrix of $\mathbf{H} = \mathbf{H} _{\text{col}} \otimes \mathbf{H} _{\text{row}}$ for each attention head where the shapes of $\mathbf{H} _{\text{col}}$ and $\mathbf{H} _{\text{row}}$ are $d \times d$ and $d _{h} \times d _{h}$, respectively ($d _{h}$ is the head dimension and $d = Hd _{h}$). Before computing the inverse Hessian, the Kronecker product of $\mathbf{H} _{\text{col}}$ and $\mathbf{H} _{\text{row}}$ ($\mathbf{H} _{\text{col}} \otimes \mathbf{H} _{\text{row}}$) needs to be computed and stored. In other words, we need to save a $d _{h}d \times d _{h}d$ matrix for each attention head, which requires more than 100 GB memory even for OPT-125M. Obviously, this is not possible with a single A100 GPU.
> >
> >  - By assuming the independence between different attention heads, we could quantize rows belonging to different attention heads simultaneously. Without such simultaneous quantization of different heads, all rows need to be quantized sequentially (respectively), which cannot properly utilize the massive compute capabilities of modern GPUs. Indeed, we check that sequential quantization of all rows results in a significantly longer (at least 10 times longer) processing time than that required by the proposed simultaneous quantization (see Table 2 in the main text).
> >
> >  - In the final version, we will discuss these points to elucidate the benefits of each proposed component.
> >
> > <List of references>
> >
> > [1] Y. Li et. al., "BRECQ: Pushing the limit of post-training quantization by block reconstruction," ICLR 2021.
> >
> > [2] W. Shao et. al., "OmniQuant: Omnidirectionally calibrated quantization for large language models," ICLR 2024.
> >
> > [3] Y. Ma et. al., "AffineQuant: Affine transformation quantization for large language models," ICLR 2024.

---

> ### Comment · Reviewer_gH6w · 2024-11-21
> **Thanks for the reply, I have raised my score.**
>
> Thanks for the thorough explanation, I think the rebuttal has addressed my concern. Also after referring the response to other reviewers, I think the rebuttal is also convincing. So I decide to raise the Contribution to 3, Confidence to 4 and Overall rating to 6.

---

> > ### Author Response · Authors · 2024-11-22
> >
> > Dear Reviewer gH6w,
> >
> > We are very glad to hear that your concerns have been addressed!
> >
> > We sincerely appreciate your recognition of our efforts and the time you dedicated to reviewing our paper.
> >
> > Respectfully,
> >
> > Authors of Paper 3463

---

### Author Response · Authors · 2024-11-21
**Look forward to further discussions**

Dear reviewers,

We sincerely appreciate the reviewers' valuable feedback and constructive suggestions.

We have made a thorough effort to address all the concerns raised.

Below is a summary of the key points in our responses.

**1. Additional Experimental Results**

 - We have included
   - results for recent language models such as LLaMA2 and LLaMA3
   - integration results with the recent transformation method QuaRot [1]
   - group-wise quantization results
 - Overall, we have achieved **at least 8% improvement** in accuracy across almost all quantization configurations, which we believe is a significant advancement.

**2. Emphasis on Our Contribution**
 - While two reviewers acknowledged the innovation of our method, two others expressed concerns about its novelty.
 - The core novelty of our work lies in capturing inter-layer dependencies **without relying on backpropagation**, unlike conventional methods (e.g., BRECQ [2], OmniQuant [3], AffineQuant [4]) that depend on time-intensive, gradient-based optimization.

**3. Comprehensive Comparisons of Quantization Processing Times**
 - We have compared the quantization processing times of the proposed method, the conventional training-free method (GPTQ [5]), and training-based methods (OmniQuant [3] and AffineQuant [4]).
 - Compared to existing training-based approaches, the proposed method enables **faster** quantization while delivering **significantly better** performance.
 - Although the proposed method requires more processing time than GPTQ due to the consideration of inter-layer dependencies, this consideration results in significantly improved performance, with **over 8% improvement**.

For more details, we kindly invite the reviewers to refer to our point-by-point responses.

Once again, we are grateful to the reviewers for their time to read and review our paper, and we look forward to further constructive discussions.

[1] S. Ashkboos et. al., "QuaRot: Outlier-free 4-bit inference in rotated LLMs," NeurIPS 2024.

[2] Y. Li et. al., "BRECQ: Pushing the limit of post-training quantization by block reconstruction," ICLR 2021.

[3] W. Shao et. al., "OmniQuant: Omnidirectionally calibrated quantization for large language models," ICLR 2024.

[4] Y. Ma et. al., "AffineQuant: Affine transformation quantization for large language models," ICLR 2024.

[5] E. Frantar et. al., "GPTQ: Accurate post-training quantization for generative pre-trained Transformers," ICLR 2023.

---

### Meta-Review · Area_Chair_FSL8 · 2024-12-20

**Metareview:**

This paper introduces BOA, a post-training quantization (PTQ) method for large language models (LLMs) that avoids backpropagation by leveraging attention-aware Hessian matrices to capture inter-layer dependencies within the attention module. BOA demonstrates improved quantization accuracy, particularly at low bit-widths (e.g., INT2), and incorporates techniques like Hessian relaxation and head-wise simultaneous quantization to reduce computational overhead. While reviewers commend the paper's focus on resource-efficient quantization for LLMs and its compatibility with other techniques like SmoothQuant and Z-FOLD, they find the contributions incremental and the novelty limited. The experiments are primarily conducted on outdated models (e.g., BLOOM, LLaMA1, OPT) and lack validation on state-of-the-art models or comparisons with recent quantization methods (e.g., QuaRot, SpinQuant). Additionally, BOA’s processing and memory overheads remain higher than some existing methods, while its performance improvements are marginal. Overall, reviewers recognize the importance of the topic but suggest that stronger experimental validation and comparisons are needed to make BOA a more compelling contribution.

**Additional Comments On Reviewer Discussion:**

The reviewer believed that the paper's positioning and contributions remain unconvincing, and he authors overlook critical benchmarks by focusing comparisons primarily on GPTQ while distancing itself from established PTQ methods like SpinQuant, AffineQuant, and OmniQuant.

---

### Decision · Program_Chairs · 2025-01-22

Reject